# Challenging Mimickers in the Diagnosis of Sarcoidosis: A Case Study

**DOI:** 10.3390/diagnostics11071240

**Published:** 2021-07-12

**Authors:** Thomas El Jammal, Yvan Jamilloux, Mathieu Gerfaud-Valentin, Gaëlle Richard-Colmant, Emmanuelle Weber, Arthur Bert, Géraldine Androdias, Pascal Sève

**Affiliations:** 1Department of Internal Medicine, Lyon University Hospital, 69004 Lyon, France; thomas_3901@hotmail.fr (T.E.J.); yvan.jamilloux@chu-lyon.fr (Y.J.); mathieu.gerfaud-valentin@chu-lyon.fr (M.G.-V.); gaelle.richard-colmant@chu-lyon.fr (G.R.-C.); emmanuelle.weber@chu-lyon.fr (E.W.); arthur.bert@chu-lyon.fr (A.B.); 2Department of Neurology, Service Sclérose en Plaques, Pathologies de la Myéline et Neuro-Inflammation, Hôpital Neurologique Pierre Wertheimer, Lyon University Hospital, F-69677 Bron, France; geraldine.androdias-condemine@chu-lyon.fr; 3Research on Healthcare Performance (RESHAPE), INSERM U1290, 69373 Lyon, France

**Keywords:** sarcoidosis, diagnosis, lymphoma, opportunistic infections

## Abstract

Sarcoidosis is a systemic granulomatous disease of unknown cause characterized by a wide variety of presentations. Its diagnosis is based on three major criteria: a clinical presentation compatible with sarcoidosis, the presence of non-necrotizing granulomatous inflammation in one or more tissue samples, and the exclusion of alternative causes of granulomatous disease. Many conditions may mimic a sarcoid-like granulomatous reaction. These conditions include infections, neoplasms, immunodeficiencies, and drug-induced diseases. Moreover, patients with sarcoidosis are at risk of developing opportunistic infections or lymphoma. Reliably confirming the diagnosis of sarcoidosis and better identifying new events are major clinical problems in daily practice. To address such issues, we present seven emblematic cases, seen in our department, over a ten-year period along with a literature review about case reports of conditions misdiagnosed as sarcoidosis.

## 1. Introduction

Sarcoidosis was first described by Jonathan Hutchinson, an English physician, in 1877 [1]. This multi-systemic disease is characterized by the infiltration of various tissues by non-necrotizing granulomas [2]. Although the mechanisms of granuloma formation are more and more clearly understood, there is currently no known cause of sarcoidosis (if there is any). Sarcoidosis can affect a wide variety of people, from the youngest to the oldest, regardless of ethnicity. Nevertheless, sarcoidosis is more common in young adults and classically starts earlier in men than women. In total, 70% of patients are aged between 25 and 40 years at presentation. A second peak of incidence is observed in women over 50 years old [3]. Its annual incidence is estimated between 2.3 and 11/100,000 while the estimated prevalence varies from 2 to 160 cases per 100,000 individuals. This large range for prevalence relies on the variability in diagnostic tools between studies as well as differences in the relative proportions of different ethnicities [4]. About two thirds of sarcoidosis patients experience a self-remitting disease without immunosuppressants, while the remaining third evolve through a chronic disease in which corticosteroids are the cornerstone of the treatment [4,5,6].

The diagnosis of sarcoidosis is still based on a set of arguments represented by three major criteria: clinical compatibility with the diagnosis of sarcoidosis, the absence of a differential diagnosis that is at least as likely, and the demonstration of epithelioid granulomas on histology [6]. Indeed, sarcoidosis can mimic other conditions which, if misdiagnosed as sarcoidosis, may result in a delay in diagnosis or treatment that is harmful to the patient especially in case of neoplastic or infectious diseases. These conditions include infections (e.g., tuberculosis, *Bartonella* spp.), neoplastic disorders (e.g., lymphomas), systemic diseases with granulomatosis (e.g., granulomatosis with polyangiitis (GPA), Crohn’s disease [CD]), common variable immunodeficiency (CVID) and drug-induced (DI) sarcoid-like reactions (SLR). Before starting invasive investigations or initiating an immunosuppressive treatment, the clinician has to pay attention to these mimickers. Moreover, sarcoidosis patients are at increased risk of developing infections [7] and more specifically opportunistic infections which could mimic sarcoidosis or imitate a sarcoidosis flare [8]. The risk of developing solid neoplasia or hematological malignancies is increased in sarcoidosis patients [9]. This statement emphasizes the need for the clinician to pay attention to these differentials in that a delayed diagnosis can have negative consequences for the patient.

We hereby describe selected cases that came to our attention after referral to our internal medicine department for a suspected sarcoidosis diagnosis. In all cases, the physical examination, radiological examinations and the advanced microbiological or histological approaches allowed the identification of these differentials. We have chosen to focus on seven sarcoid mimickers and to summarize their main clinical features in light of their misleading or atypical presentation and the associated diagnostic approach.

## 2. Materials and Methods

This article is a series of original cases and adheres to the principles of the Declaration of Helsinki of 1964 and its latest amendments. The informed consent of all participants was obtained, and the study was approved by the Local Institutional Review Board. The case reports were chosen among consecutive patients referred to our internal medicine department. Each case report was selected as it was considered by the authors to be representative of the diagnostic issues faced by the clinician. This study received approval from the local ethics committee in February 2019 (No 19–31). In order to describe the updated diagnosis of these diseases, we conducted a review of the English and French medical literature on the Medline database, using the keywords “sarcoidosis”, “mimickers”, “misdiagnosed” and “differential diagnosis”. We excluded articles written in languages other than English or French.

## 3. Clinical Cases

### 3.1. Infectious Diseases

#### 3.1.1. Case Description

A 70-year-old man was referred to our internal medicine department for suspected oral sarcoidosis. He had a past medical history of type 2 diabetes treated with insulin, and prostate cancer in remission treated with radical prostatectomy along with adjuvant radiation therapy. He reported a trip to India four months after the onset of symptoms, with no medical issues at the time.

He noticed the appearance of an ulceration around tooth 38 in June 2016. A first gingival biopsy was performed by his dentist three months later, finding giant-cell granuloma without necrosis. A second biopsy was performed in a maxillofacial surgery department, showing again a non-necrotizing giant-cell granuloma pattern. Neither sample was tested for mycobacteria (Ziehl Nielsen staining, culture or polymerase chain reaction [PCR]). Of note, a few days after returning from India (a trip the patient took after the onset of oral ulceration), he developed dry cough which led to further investigations. A thoracic CT scan showed interstitial lung disease with diffuse micronodules, retractive consolidations, and traction bronchiectasis. There was no mediastino-hilar adenopathy. This pattern was initially considered to be consistent with thoracic sarcoidosis. Oral corticosteroid therapy 40 mg/day was then started, which improved the dry cough, but with significant corticosteroid dependence and corticosteroids side effects such as decompensation of his diabetes. The patient was then referred to our internal medicine department for further management of his atypical, presumed sarcoidosis.

Detailed patient interview did not reveal any recent altered general condition nor extra-respiratory symptoms. A magnetic resonance imaging (MRI) of the jaw was performed and evidenced a left mandibular lesion extending to the cheek, with numerous satellite enlarged lymph nodes, including a necrotic one. Biological examinations revealed an inflammatory syndrome with an elevated C-reactive protein (16.9 mg/L), and a positive interferon-gamma release assay (TB GOLD TEST QUANTIFERON: 5.66). There was no elevation of angiotensin-converting enzyme levels. A bronchial fibroscopy was performed. The bronchial biopsy evidenced nothing except aspecific inflammation without granuloma.

Finally, the bronchoalveolar lavage fluid culture revealed the presence of *Mycobacterium tuberculosis* (Mtb). The search for *rpoB* mutations was positive, indicating a rifampicin resistant Mtb strain. The complete sequencing of the strain revealed mutations in favor of additional resistance to ethambutol, pyrazinamide, and ethionamide, defining extensively drug-resistant tuberculosis. The Mtb PCR performed a posteriori on the 2nd oral biopsy was positive, indicating that tuberculosis was contracted before the trip to India. New ear nose and throat (ENT) examination with laryngoscopy revealed a pharyngeal lesion extending to the epiglottis, in favor of laryngeal tuberculosis.

The patient was transferred to the infectious diseases department and a combined antibiotic therapy of linezolid, amikacin, levofloxacin, cycloserin, bedaquillin, and para-aminosalicylic acid was initiated. Linezolid was stopped after 6 months because of neurotoxicity, as was amikacin, given the positive evolution. Other antibiotics were maintained for a total treatment duration of 18 months. ENT lesions gradually disappeared and the control BALF culture at 2 months of treatment was sterile.

#### 3.1.2. Discussion

Non-caseating giant cell granulomas are the histological hallmark of sarcoidosis. Unfortunately, these are not specific to any disease and many infectious disorders may present with granulomatous features (Table 1). Tuberculosis is one of the main differential diagnoses of sarcoidosis especially in case of lung and lymph node involvement. Tuberculosis is worldwide way more frequent than sarcoidosis in that in 2019, 10 million people were affected by tuberculosis across the world.

Tuberculosis and sarcoidosis are hard to distinguish from each other since even extra pulmonary manifestations may be similar in both diseases (i.e., uveitis, digestive tract involvement, neurological involvement, peripheral lymphadenopathies, arthritis, hypodermitis). Several general and nonspecific clinical signs may be present in both diseases, including fever, weight loss, night sweats, malaise, and fatigue, as well as being born abroad in a country where tuberculosis is highly endemic and having been in contact with tuberculosis [58]. Of note, some of the patient’s history features can be atypical for sarcoidosis, notably hemoptysis and centrolobular micronodules, cavitary lesions and unilateral adenopathy. In case of ocular involvement, occlusive vasculitis, serpiginous-like choroiditis, single choroidal granuloma and perivascular choroiditis patches are more consistent with tuberculosis whereas segmental and nodular vasculitis are more prone to revealing sarcoidosis [59]. ENT involvement is not rare in sarcoidosis patients but oral involvement (e.g., gingival and jugal involvement) is atypical and should raise the question of a differential diagnosis.

Various diagnostic tools are available for mycobacterial infections. Tuberculin skin tests can be falsely negative in sarcoidosis patients defining tuberculin anergy. Interferon gamma release assays (IGRAs) are valuable alternatives in this setting to diagnose latent tuberculous infections (LTBI). This type of test is not applicable to the diagnosis of active TB due to the non-neglectable amount of false negative results in active-TB patients [60].

Sputum microscopy searching for Mtb acid-fast bacilli and liquid phase cultures are the most widely used tests for tuberculosis diagnosis [61]. In recent decades, the advances of molecular genetics have allowed for an improvement in the detection of Mtb and antibiotics resistance of the strains. Several molecular detection tests for Mycobacterium tuberculosis relying on polymerase chain reaction are available. The Xpert MTB/RIF test is probably the most known of rapid molecular tests for tuberculosis diagnosis. Its principle is based on heminested PCR assay to amplify sequences from the *rpoB* genes with specific probes to target mutations within the rifampicin-resistance determining region [62]. The sensitivity and specificity of the test rely on several pre-test conditions including smear positivity, HIV status, sample site (e.g., pulmonary or extra pulmonary), age and endemic burden in the country [63].

Several studies found that search for genetic material from tuberculous and non-tuberculous mycobacterial (*M. paratuberculosis*, *M. avium*) in tissues from sarcoidosis patients could be positive despite the absence of any infectious sign [64,65,66]. One assumes that several environmental triggers, including mycobacterial antigens, could promote the formation of granulomas in predisposed individuals. Moreover, several animal models of granuloma are induced through mycobacterial derived antigens (either from Mtb or non-tuberculous mycobacteria) [67].

Other infectious conditions can present as granulomatosis. When facing a diagnosis of granulomatosis, the first step is to assess which organ is affected to perform a correct differential diagnosis [13]. If granulomas are found within the liver, hepatitis C or B viruses must be ruled out. Fibrin-ring granulomas within the liver should prompt a search for Q fever, cytomegalovirus (CMV) or Epstein–Barr Virus (EBV) [68]. When associated with eosinophil granulocytes, granulomas should invoke the diagnosis of schistosomiasis, especially in endemic regions [13]. In endemic areas, *Histoplasma capsulatum* can present with granulomatous lung disease along with bone marrow, liver, spleen and lymph node involvement while adrenal glands, skin and mucosal involvement are less frequent [69].

Thus, when facing an unexplained granulomatosis, infectious differential diagnoses should be ruled out (especially tuberculosis). Bacterial, fungal, viral and parasitological investigations should be performed, either systematically for tuberculosis or according to the patient’s origin, previous trips in endemic areas or according to the specific organ involvement pattern (Table 1).

A careful histological examination is also a key point for an accurate diagnosis since some pathological features help to orientate a specific disease. Tuberculoid granuloma with central caseous necrosis helps to orientate mycobacterial infections, cryptococcosis, histoplasmosis and brucellosis, while suppurative or pyoepithelioid granulomas help to orientate bartonellosis, yersisniosis, chlamydiosis or tularemia [70]. Whipple’s disease granulomas, as well as syphilitic granulomas, more frequently present with a sarcoid-like pattern. When histological examination reveals central necrosis, one should focus on the type of necrosis. Fibrinoid necrosis is different from caseous necrosis with an intense red color marking fibrin on hematoxylin eosin staining [71]. This pattern is associated with vascular damage more frequently linked to vasculitis either immune-mediated vasculitis (e.g., rheumatoid arthritis, ANCA associated-vasculitis [AAV]) or infectious vasculitis (e.g., rickettsiosis, syphilis).

### 3.2. Neoplastic Disorders

#### 3.2.1. Case Description

A 66-year-old male patient was admitted to our internal medicine department to explore mild asthenia, with a 2 kg weight loss, diffuse myalgia and persistent biological inflammatory syndrome (C reactive protein ranging from 100 and 220 mg/L). His medical history included diffuse cardiovascular disease, duodenal ulcer and resected bladder polyps.

The symptoms had been present for two months. The physical examination was unremarkable. Chest X-ray and abdominal ultrasound were initially considered normal. Initial laboratory workup showed no cytopenia, no renal or hepatic dysfunction, no hypercalcemia, lactate dehydrogenase was slightly elevated (284 UI/L). Blood and urine cultures, Quantiferon, HIV, hepatitis B and C serology were negative. The Angiotensin conversion enzyme dosage was within the normal range (36.9 U/L). A Thoracic and abdominal CT found multiples mediastinal adenopathies (the largest in the Barety’s space measuring 23 mm), and cervical adenopathies, with several aspecific pulmonary micronodules. Since the first line investigations remained undiagnostic, a bone marrow aspiration was performed. The myelogram and blood clonality found no evidence of hemopathy. Broncho alveolar lavage fluid culture found no microbial agent, including *Mycobacterium*
*tuberculosis*, but a lymphocytic alveolitis (90%) with CD4/CD8 ratio of 1.27.

The biopsy of a cervical adenopathy found pyoepithelioid inflammation, with partial necrosis and histiocytes, without giant cell. A broader infectious research was then realized: universal PCR, *Tropheryma whipplei* specific PCR and cultures on the adenopathy were negative. *Yersinia* spp., *Bartonella* spp., *Francisella tularensis*, *Chlamydia* spp. and syphilis serology were negative.

In case of occult infection with intracellular bacteria, a probabilistic treatment with doxycycline for one month was implemented without improvement, with the appearance of intense pruritus and bullous skin lesions located on the extremities. The clinical picture was suggestive of an inflammatory disease (enlarged lymph nodes, skin lesion, pruritus, persistent fever and inflammatory syndrome) and treatment with corticosteroids was discussed. However, new symptoms (mild fever, polyneuropathy) appeared before it could be started, which led to a second workup. A positron emission tomography (PET)-CT showed growing mediastinal adenopathies (largest measured at 32 mm), which were hypermetabolic (standardized uptake value: 5). Electroneuromyography found diffuse axonal polyneuropathy, lumbar puncture (mild hyperproteinorachia 0.7 g/L, no cells, no neoplastic infiltration, no oligoclonal bands) and brain MRI was unremarkable. Finally, a biopsy of the mediastinal adenopathy found Hodgkin’s disease of scleronodular subtype. The evolution was favorable after chemotherapy.

#### 3.2.2. Discussion

Neoplasia are another of the great mimickers of sarcoidosis. In cancer patients, systemic granulomatosis and SLR associated with cancer can occur either before, during or after the onset of the neoplasia. These granulomatous disorders can occur in many conditions in cancer patients (i.e., related to opportunistic infections associated with cancer, systemic granulomatosis due to cancer treatment, especially checkpoint inhibitors or BRAF/MEK inhibitors) [9].

Apart from the increased risk of cancer in sarcoidosis patients, SLR to tumor cells can be encountered in various types of cancer, either in solid or hematological malignancies [72]. Some authors have suggested that satellite SLR to cancer could be the marker of an efficient immune response to tumoral cells and thus making it a marker of good prognosis [73,74,75,76]. Systemic granulomatosis in the context of cancer can also occur. Arish et al. described a series of 29 patients diagnosed with sarcoidosis in a context of preexisting cancer. Breast cancer and lymphoma were the most frequently observed cancer types. These patients presented features of systemic granulomatosis (lymphocytic alveolitis, mediastinal and hilar lymphadenopathies, endobronchial granulomas in histological examination) but most of them were strictly asymptomatic [77].

Our case illustrates the need both for the pathologist to examine the specimen carefully but also for the clinician to raise awareness of the search for a particular disease, and in particular Hodgkin’s disease in the context of atypical granulomatosis. In our case, Hodgkin’s disease was associated with an important inflammatory syndrome and a non-specific chest CT pattern for sarcoidosis. These atypical features lead to a second diagnostic workup allowing a correct diagnosis at the end. Interestingly, another atypical feature was the pyoepithelioid inflammation which is unusual in sarcoidosis. In a recent series of necrotizing granuloma-like Hodgkin’s lymphomas (NGHL), Hou et al. identified 34 cases of NGHL in 1887 Hodgkin’s lymphomas (2%) over a period of 14 years in their tertiary center in China [78]. In their series, NGHL cases were compared with non-NGHL cases. The authors found that the NGHL subtype, although rare, is associated with a poorer prognosis. Interestingly, three cases of NGHL in this series were initially misdiagnosed as cat scratch disease.

Sarcoidosis patients have also an increased risk of neoplasia as compared with general population. In 1973, Brincker et al. described an increased risk of lymphoma in a cohort of 2544 sarcoidosis patients [79]. In 1986, the same team described the sarcoidosis-lymphoma syndrome (SLS), a condition in which sarcoidosis patients develop a lymphoproliferative disorder, mostly in their fourth decade, in a median time of 24 months after the diagnosis of sarcoidosis [80]. The most frequent subtype of lymphoma encountered in the first series of SLS from Brincker is Hodgkin’s disease whereas Papanikolaou and Sharma evidenced that non-Hodgkin’s lymphoma was more frequent than Hodgkin’s lymphoma [9]. In this subgroup, patients are usually older and present B symptoms more frequently.

Another subtype of lymphoproliferative disease mimicking sarcoidosis is lymphomatoid granulomatosis (LYG). LYG is a lymphoproliferative EBV-associated clonal disorder usually presenting with lung, lymph nodes, central nervous system and skin involvement in immunocompromised patients [38]. The typical histological patterns is represented by the coexistence of granulomatous inflammatory features composed of atypical and EBV positive CD20 positive B cells along with lymphocytic vasculitis and a certain amount of necrosis [81]. Its aggressiveness potential varies widely, from that of a diffuse large cell B lymphoma to that of an indolent lymphoma. An aggressive lymphoma occurs in 7 to 47% of LYG cases and LYG sometimes shares aggressive features with lymphoma thus indicating the need for chemotherapy [82]. This example enhances the necessity of a full examination of all samples when facing an atypical granulomatosis (i.e., with deeply impaired general condition or primary corticosteroid resistance).

Finally, in hematological malignancies, a specific subtype of sarcoidosis called “donor-acquired sarcoidosis” can occur. This represents the putative transmission of sarcoidosis from a bone marrow donor to its recipient [83,84,85]. Moreover, unregarding the donor medical history, bone marrow transplant recipients for hematological malignancies can develop sarcoidosis either in a context of allogenic [84,86,87] or autologous [88,89] stem cell transplant.

### 3.3. Inflammatory Disorders and Systemic Diseases

#### 3.3.1. Case Description

A 40-year-old female was admitted to the emergency department for one-month lasting abdominal symptoms consisting of severe abdominal pain which was relieved by diarrhea. She reported severe fatigue and a 5 kg weight loss. She had taken non-steroidal anti-inflammatory drugs for a dental abscess seven days earlier. She had low grade fever (38.4 °C). The patient had a history of iron deficiency anemia, a cesarean section and a flange digestive occlusion. On physical examination, the abdomen was tender throughout without hepatosplenomegaly or palpable lymph nodes. Otherwise, the physical examination was unremarkable.

Blood tests found microcytic anemia (Hemoglobin (Hb): 9 g/dL, mean corpuscular volume of 80 fl) with both iron and vitamin B12 deficiency (ferritin: 12 µg/L, B12 93 pmol/L (138–652 pmol/L)) and elevated levels of acute phase reactants (i.e., C-reactive protein (CRP): 45 mg/L). Renal and liver function tests were both normal.

The gynecological examination with endo-vaginal ultrasound was normal. Chest X-ray was also normal. The abdomino-pelvic CT scan showed ileitis associated with multiple mesenteric lymph nodes enlargement up to 12 mm. Hepatosplenomegaly of 12 cm with multiple intraparenchymal hypodense nodules was also described. Neither pathogens including tuberculosis (culture and PCR [intestine samples]), toxoplasmosis (serology), histoplasmosis (serology) and leishmaniasis (serology and PCR), nor any autoantibodies were found except positive parietal cell antibodies. Serum electrophoresis was normal as was angiotensin converting enzyme dosage. The culture and parasitological examination of the stools were negative as were serological testings for HIV and syphilis. Fecal calprotectin was elevated at 1168 µg/g (N < 50 µg/g). Bone marrow aspiration evidenced a normal richness without pathological infiltration but with mild signs of dysmyelopoiesis. There was no macrophagic overload, especially Gaucher cells.

The diagnostic work-up was completed with a splenic MRI that showed multiple solid splenic lesions (Figure 1). Gastroscopy and ileocolonoscopy showed a nippled appearance of the terminal ileum without any other notable abnormality. Histological examination of the digestive tract biopsies revealed an atrophic gastritis with intestinal metaplasia compatible with Biermer disease. A PET-CT was performed and showed hypermetabolism of the splenic nodules, ileum, mediastinal and right supra clavicular lymph nodes, as well as pulmonary micronodules (Figure 2). Pulmonary function tests showed normal respiratory volumes without ventilatory or diffusion disorder. Bronchoscopy showed a normal macroscopic aspect of the respiratory tract. Bronchoalveolar lavage fluid (BALF) analysis evidenced a lymphocytic alveolitis with an increased CD4/CD8 ratio (13.34, N: 0.6–5.50). Infectious samples were negative, especially the mycobacterial cultures. Histological examination of bronchial biopsies showed a non-caseating giant cell granuloma suggesting a diagnosis of sarcoidosis.

The clinical picture was initially suggestive of systemic sarcoidosis with diffuse mediastinal, supra- and subdiaphragmatic lymph node involvement, associated with a multinodular spleen and pulmonary parenchymal involvement with bilateral micro-nodules. A corticosteroid treatment at 0.7 mg/kg per day was started regarding the major splenic and lymph node involvement responsible for impaired general condition.

The patient was then referred to our department. Because of the initial presentation with abdominal pains and ileitis, persistent digestive disorders under corticosteroids, we considered the diagnosis of Crohn’s disease (CD) and cobalamin deficiency related to ileum involvement. A capsule endoscopy was performed and revealed multiple ulcerated lesions of the small intestine with two large ulcerations of 30 to 50% progression. The diagnosis of CD was finally retained. Infliximab 5 mg/kg was started in association with the decrease of corticosteroid therapy which allowed the rapid improvement of the patient’s general condition and digestive symptoms.

#### 3.3.2. Discussion

Several inflammatory or systemic disorders can mimic sarcoidosis or present with granulomas (Table 2). Granulomas can be found on histological samples in patients with anti-neutrophil cytoplasmic antibodies (ANCA)-associated vasculitides (AAV) either GPA or eosinophilic granulomatosis with polyangiitis (EGPA) and several presentations (e.g., crusting rhinitis, destructive sinusitis and orbital pseudotumor, myocarditis, arthritis as well as neurological manifestations) can be misdiagnosed as sarcoidosis [90].

Granulomatous lung diseases are a heterogeneous group of diseases including, in addition to sarcoidosis and granulomatous AAV, both infectious diseases (e.g., mycobacterial or fungal infections) and non-infectious diseases (e.g., hypersensitivity pneumonitis, pneumoconiosis, nodular rheumatoid arthritis, Langerhans’ cell histiocytosis, and bronchocentric granulomatosis) [103]. The positivity of ANCA with anti-PR3 specificity is a core feature for GPA diagnosis, since these antibodies are found in at least 90% of patients with systemic diseases and in 75–80% of patients with limited GPA without renal involvement, which was not the case in the patient described above [104].

Necrotizing sarcoid granulomatosis (NSG) is a rare granulomatous disorder of the lung associated with vasculitis, which could be discussed as an alternative diagnosis in our patient. It is still controversial whether it is a discrete entity or a variant of nodular sarcoidosis. Its main features include (1) histological sarcoidosis-like granulomas, granulomatous vasculitis and a variable amount of necrosis, usually coagulative and caseous, (2) radiological multiple lung nodules without hilar lymphadenopathy, and (3) a benign clinical course. The clinical symptoms of NSG are often nonspecific (e.g., fever, chest pain, weight loss, cough and dyspnea) and the radiological findings vary widely (e.g., bilateral nodules and masses, cavitation, and pleural effusion). NSG does not usually affect extrapulmonary organs [105].

We report above the association of CD and sarcoidosis. Such an association is considered to be exceptional [106]. In an epidemiological study conducted in the United Kingdom, Rajoyira et al. reported that no significant association could be made between sarcoidosis and CD [107]. Contrarily, Halling et al. found an association between these two diseases, however exclusively in males with CD [108].

Digestive tract involvement is extremely rare in sarcoidosis, described in 0.1 to 1.6% of cases [109]. Even if the whole digestive tract can be involved, the stomach, and particularly the antrum, is the preferred location for digestive tract involvement in sarcoidosis [110]. The symptoms related to digestive tract involvement are usually non-specific and the diagnosis requires the presence of noncaseating granulomas within digestive tract histological samples and the exclusion other causes of granulomatous diseases including gastrointestinal-specific infections (*Salmonella* spp., *Yersinia* spp., *Campylobacter* spp., *Helicobacter* spp., *Schistosoma* spp.,…), malignancies, vasculitides, foreign material reactions (talc, starch, barium, fecal material including pulse granuloma, pneumatosis) The main alternative diagnosis of granulomatous digestive disease is CD [110], which can also be associated with extra digestive features evocative of sarcoidosis such as arthritis, erythema nodosum and uveitis [109]. As compared with CD, digestive tract sarcoidosis occurs more frequently in African Americans and is more frequently associated with weight loss and ileum sparing. During the course of the disease, patients with digestive tract sarcoidosis classically do not require surgery and clinical digestive remission is more frequent.

Nucleotide-binding oligomerization domain-containing protein 2 (*NOD2*) polymorphisms in the leucine-rich repeat domain are positively associated with CD [111]. Of note, *NOD2* mutations are present in Blau syndrome (BS), a differential diagnosis of early onset sarcoidosis [99].

Other diseases sharing the same impaired NOD2 pathway background may mimic sarcoidosis. We previously reported the case of a 19-year-old man referred to our department by ophthalmologists with suspicion of sarcoid uveitis [112]. He had a 15-year medical history of juvenile arthritis and bilateral panuveitis along with multifocal choroiditis. Given the presence of camptodactyly, the ophthalmological findings and the lack of intrathoracic adenopathies, the patient was diagnosed with BS. The complete sequencing the *NOD2* gene evidenced a heterozygous gain-of-function missense mutation (Arg334Trp) which was already described in BS [113]. BS is transmitted as an autosomal dominant trait. It is also characterized by fever, skin rash (mainly non-confluent erythematous or millimetric pigmented papules) [114] and granulomatous inflammation of the involved organs [115]. Non-erosive arthritis is almost constant and mostly polyarticular, then oligoarticular and involves wrists, ankles, knees and proximal interphalangeal joints [116]. Rosé et al. reported expended manifestations (visceral, vascular) beyond the classic clinical trial in half of the patients [116]. Physicians should consider BS as a differential diagnosis of early onset sarcoidosis in children with unusual features such as camptodactyly and normal chest computed tomography (CT), even without familial history since BS can also be sporadic [117,118].

### 3.4. Drug-Implant- and Device-Induced Sarcoidosis

#### 3.4.1. Case Description

A 68-year-old patient had metastatic melanoma of an unknown primary tumor (with BRAF V600E mutation) diagnosed in 2008, with lung, adrenal, kidney, brain and bone metastases. He had been receiving vemurafenib as a fourth line treatment since May 2011, under a temporary authorization for use (French procedure allowing its use before its marketing authorization). This treatment resulted in rapid improvement of secondary lesions. 8 months after the introduction of vemurafenib, he presented with a purplish papular skin eruption at the elbow flexion sites, and on previous scars of the chest (Figure 3).

Skin biopsy revealed numerous epithelioid and gigantocellular granulomas, without caseous necrosis, suggesting cutaneous sarcoidosis. In the following months, he presented a non-granulomatous bilateral and recurrent anterior uveitis. He was therefore referred to our internal medicine department for suspected cutaneous and ocular sarcoidosis. Angiotensin conversion enzyme (ACE) levels were increased, and the remaining laboratory examinations were unremarkable. There was no parenchymal lung involvement on chest CT. The etiologic workup of this uveitis excluded other secondary causes of uveitis. Corticosteroids eyedrops were successfully introduced. However, the uveitis became chronic with multiple flares at topical corticosteroids withdrawal, and oral corticosteroids were therefore initiated in combination with hydroxychloroquine. While the uveitis and cutaneous granulomatosis were controlled under this treatment, the patient experienced a relapse of his metastatic melanoma. Unfortunately, he died of colonic perforation due to secondary localizations of his melanoma despite the use of several unsuccessful therapeutic lines.

#### 3.4.2. Discussion

Drug-induced SLR can be defined as a “granulomatous reaction that is indistinguishable from sarcoidosis and occurs in temporal relationship with initiation of an offending drug” [119]. Many different therapeutics and microparticles were described to be responsible for SLR in the medical literature (Table 3). In addition to the reported case, we managed seven patients with DISLR due to interferon-α (*n* = 1), etanercept (*n* = 2), adalimumab (*n* = 2), tocilizumab (TCZ) (*n* = 1) and immune checkpoint inhibitors (*n* = 1) over the 4 last years.

DISLR is well described during interferon therapy, (especially with interferon-α), which is known to induce SLR along with lung involvement in up to 76% of patients with hepatitis C and granulomatosis in a retrospective study [120]. In this study, Ramos-Casals et al. found that cutaneous granulomas were diagnosed in 60% of cases.

DISLR has also been described with tumor necrosis factor inhibitors (TNFi) and especially with the soluble receptor etanercept [119]. A literature review published last year found 107 DISLR cases, including 57 cases with etanercept, 27 cases with adalimumab, 21 cases with infliximab and 7 with other TNFi, treated during a mean period of 25.6 months (ranging from 1 to 132 months) [121]. Clinical symptoms were pulmonary (51%) and cutaneous (33%). In total, 14 patients had ocular involvement followed by salivary glands (*n* = 8), kidneys (*n* = 7), central nervous system (*n* = 5), and liver (*n* = 4) involvement. TNFi were discontinued, except in two cases. All but five cases showed clinical improvement after TNFi discontinuation with (*n* = 43) or without (*n* = 41) systemic corticosteroids, drug replacement (*n* = 5) or combined therapy (*n* = 10). The mechanisms evoked to explain the paradoxical granulomatosis development during anti-TNF therapy include increased susceptibility to infections and changes in cytokine and cellular environment.

A 47-year-old woman of African origin with a past medical history of rheumatoid arthritis (RA) for 14 years was referred to our department for a mediastinal histologically proven sarcoidosis. RA was treated with TCZ which was discontinued after the onset of the SLR. RA flared concomitantly with TCZ discontinuation. Thereafter, we introduced abatacept with good efficacy and safety, while hilar and mediastinal adenopathies regressed. While few cases showed the efficacy of anti-interleukin-6 (IL-6) or IL-6 receptor (IL-6R) agents in refractory sarcoidosis [3], new-onset sarcoidosis during TCZ treatment has been described in five case reports [137,138,139,140,141]. In three cases, there was what was presumed to be TCZ-induced sarcoidosis, observed during treatment for RA, at least one year after the start of the treatment. Theodosiou et al. and Del Giorno et al. recently reported TCZ-induced sarcoidosis in patients treated for giant cell arteritis, which occurred, respectively 12 and 8 months after TCZ initiation. In all these cases, the skin was involved as the main clinical feature, associated with bilateral hilar lymphadenopathy in three patients.

Immune checkpoint inhibitors, such as anti-cytotoxic T-lymphocyte-associated protein-4 (CTLA-4) antibodies (ipilimumab) and anti- programmed cell death protein 1 (PD1) (nivolumab, pembrolizumab) or anti-PDL1 (ligand) antibodies (atezolizumab, durvalumab, avelumab) can trigger DISLR [122]. BRAF and MEK inhibitors (dabrafenib, vemurafenib and trametinib, cobimetinib) were also associated with SLR in 37 patients in the World Health Organization database Vigibase. Like what is observed with other DISLR, immunotherapy generally triggers paucisymptomatic lung, skin and lymph nodes involvement (particularly mediastino hilar involvement). The onset of such manifestations varies between 1 and 22 months with a mean of 6 months after the initiation of treatment [142]. SLR can make one wrongly evoke cancer progression, and can include lung and liver nodules, localized or diffuse adenopathies and bone lesions [143]. Kim et al. recently reported a series of 32 patients with cancer who had a history of sarcoidosis and received immune checkpoint inhibitors [144]. Of these, only one patient had a sarcoidosis flare which required treatment. These findings suggest that most patients affected with cancer can receive immune checkpoint inhibitors without experiencing sarcoidosis flares, even if they have a past medical history of sarcoidosis.

Apart from occupational or environmental diseases (Table 3) which present with granulomatous interstitial lung disease, SLR has been associated with inorganic particles, in a few reports, after joint replacement surgery [145] and after silicone breast implant placement [136]. These reactions are postulated to be due to an autoimmune inflammatory syndrome induced by adjuvants (ASIA) which existence was postulated by Shoenfeld et al. in 2011 [146]. The inability of the immune system to get rid of these antigens could promote a chronic inflammatory response which can enhance the antigen exposure to antigen-presenting cells [147]. This clinical entity should be differentiated from siliconoma which is exclusively associated with silica deposits.

We recently reported the case of a 44 year old woman of Caribbean origin referred to our department in a context of anterior bilateral granulomatous uveitis, interstitial lung disease, bilateral axillary lymphadenopathy and paresthesia along with neuropathic pain related to multineuritis [135]. She had a medical history of augmentation mammoplasty for esthetic purposes eight years ago, with a recent surgically treated breast implants rupture associated with siliconomas within breasts and enlarged axillary lymph nodes. Transbronchial, axillary lymph node and fibular nerve biopsies evidenced granulomatous inflammation. Further investigations with electron microscopy and energy dispersive X-ray spectrometry were performed on transbronchial, lymphatic and nerve biopsies which revealed silicon particles within all the samples except nerve biopsy. We retained the final diagnosis of systemic granulomatosis linked to silicone spread and started corticosteroids with hydroxychloroquine, with good clinical efficacy. Overall, this observation suggests the interest of scanning electron microscopy and energy dispersive X-ray spectrometry for the diagnosis of implant and device-induced sarcoidosis.

### 3.5. Primary Immune Deficiencies

#### 3.5.1. Case Description

A 53-year-old female patient of Portuguese origin was admitted to our internal medicine department in the context of a one-month history of right supra-clavicular adenopathy. She reported a 10 kg weight loss with asthenia for the past six months. She had a past medical history of meningitis, repeated cystitis and left cervicobrachial neuralgia. She also reported a history of stomach cancer in her mother and colon cancer in her brother.

The initial laboratory work-up showed isolated lymphopenia at 0.92 G/L, a moderate inflammatory syndrome and a mild hypercalcemia (2.7 mmol/L). Serum protein electrophoresis showed normal gammaglobulin count (7 g/L). Quantiferon was negative, angiotensin converting enzyme was increased to 119 IU/L and lysozyme to 59.2 IU/L. Cervical ultrasonography and thoracic and abdominal CT scans showed a right supra-clavicular adenomegaly (25 mm) with mediastinal and hilar nodes and a pulmonary infiltrative syndrome with bibasal reticulations, bronchiectasis and ground glass opacities. Lung function tests showed minimal distal bronchial obstruction. There was no restrictive ventilatory disorder and no decrease in alveolar-capillary diffusion. Bronchial fibroscopy showed macroscopic granulomatous infiltration with an increased CD4/CD8 ratio (5.74). Bronchial biopsies showed epithelioid and gigantocellular inflammation without caseous necrosis. Tuberculin skin test (TST) was negative. The supra-clavicular lymph node microbiopsy also showed epithelioid and gigantocellular granulomatous inflammation without caseous necrosis. Electrocardiography was normal with good left ventricular systolic and diastolic functions. The diagnosis of systemic sarcoidosis was made, and corticosteroids were started at 0.5 mg/kg/day for one month with a progressive withdrawal.

The initial clinical course was favorable with partial recovery of the initial weight and regression of enlarged lymph nodes. The thoracic CT showed the persistence of sequelae of fibrotic lesions. However, she subsequently developed hypercalcemia partly related to the discovery of a parathyroid adenoma, leading to the addition of hydroxychloroquine 400 mg per day, a further increase in corticosteroids dosage and secondary with surgical management of the parathyroid adenoma. She also presented with right peripheral facial palsy with monoparesis of the right upper limb, which rapidly resolved under 1 mg/kg/day of corticosteroid. Neurological investigations (lumbar puncture and brain MRI) were normal without any sign suggestive of neurosarcoidosis. The corticosteroid therapy was stopped after one year of treatment and hydroxychloroquine was stopped one year later.

A new serum protein electrophoresis showed persistent hypogammaglobulinemia at 4 g/L with IgG deficiency at 4.37 g/L (7–16 g/L), IgA at 0.62 g/L (0.7–4 g/L) and IgM at 0.41 (0.4–3 g/L). Lymphocytes immunophenotyping confirmed a decrease in memory B cells (switched and non-switched) and an increase in naive B cells in favor of a common variable immunodeficiency (CVID) diagnosis. There was no transitional B cells expansion. There was a slight decrease of T cells (359/µL) along with low CD4 count (205/µL) under corticosteroids. Activated T cell phenotyping was unremarkable. The patient remains well under immunoglobulin substitution.

#### 3.5.2. Discussion

The concept of immune deficiency has recently undergone major changes with advances in genetics. Thus, the group of primary immunodeficiencies, which is vast and heterogeneous, is now merged with that of inborn errors of innate immunity (IEIs). In 2021, there have been more than 450 single-gene innate errors of immunity underlying phenotypes encompassing as diverse manifestations as infections, malignancies, allergy, autoimmunity [148]. These phenotypes may be exclusive, but are most often overlapping, with immune deficiency being combined with, for example, autoimmune manifestations or lymphoproliferation [149]. The identification of genetic abnormalities, in addition to providing pathophysiological clues, also opens the way to the development of targeted specific treatments.

CVID, the most frequent primary immune deficiency in adults with an estimated prevalence worldwide between 1/10,000 and 1/100,000 [150], is now linked to anomalies in more than 20 genes [148,149]. Of note, CVID has a familial aggregation in 5–25% of the cases, and a single gene mutation is identified in less than 10% of patients [151]. In its original definition, CVID is an increased susceptibility to infections (particularly encapsulated bacteria) linked to a defective antibody production in the more global context of hypogammaglobulinemia. According to the European Society for Immuno deficiencies, CVID results from a decreased immunoglobulin (Ig) G level (<2 standard deviation below the mean for age) associated with a drop in at least another Ig isotype (IgA or IgM) or in IgG subpopulations (such as IgG2 or IgG4). This also results in a decreased response to vaccination whether it consists of a T dependent or T independent antigen presentation.

Immunophenotyping may also be indicative of CVID when it shows a decrease in switched memory B cells and in whole memory B cells population (CD19+CD27+IgD-IgM- and CD19+CD27+IgD+). During the diagnostic workup, the clinician should focus on ruling out secondary hypogammaglobulinemia (e.g., myeloma, chronic lymphocytic leukemia, cryoglobulin, Good’s syndrome, nephrotic syndrome or exudative enteropathy, and drug-induced hypogammaglobulinemia).

A rarer form of combined immunodeficiency affecting older adults, the Late-Onset Combined Immunodeficiency (LOCID, 5–8% of CVID cohorts) is characterized by CD4 lymphopenia (naive T cells) and an increased susceptibility to opportunistic infections [152].

In addition to being associated with increased risk of autoimmune manifestations (relative risk of autoimmune cytopenias >100), increased risk of cancer (lymphoma, gastric cancer), CVID predisposes to lymphoproliferation, either as lymphoid hyperplasia (4–7%) or granulomatosis (9–15%), which can mimic a full-blown sarcoidosis [153,154]. In other cases (2–6%), lymphoproliferation is malignant thus the differential diagnosis with lymphoma may be difficult and requires (sometimes repeated and orientated/extensive) histological analysis.

Granulomatosis may precede or reveal the diagnosis of CVID. In other cases, the diagnostic criteria for CVID may be absent at the onset of the disease and lowered immunoglobulin levels may occur secondarily. It is therefore essential to perform serum protein electrophoresis before concluding the diagnosis of sarcoidosis. Sarcoidosis is associated with hypergammaglobulinemia in at least half cases [155,156] and normal electrophoresis (or low borderline values but not below normal values) should alert the clinician. Immunoglobulin subclass assays and lymphocyte immunophenotyping will then be helpful.

Some clues may point to CVID rather than sarcoidosis, such as a familial aggregation (or consanguinity), past medical history of repeated infections, concurrent autoimmune manifestations, atypical presentation, predominantly extra-pulmonary involvement, diffuse and multi-organ involvement (especially involvement of the spleen), or lymphoid hyperplasia involving extra-pulmonary organs. In chest X-rays or computed tomography (CT) scans, CVID-associated pulmonary involvement (whether it is granulomatous lymphocytic interstitial lung disease or lymphocytic interstitial pneumonia) is often more severe, with a higher frequency of random nodules over micronodules (with a lesser perilymphatic distribution), more frequent halo signs, bronchiectasia and less frequent hilar adenopathies compared with sarcoidosis patients. Finally, the CD4/CD8 ratio in BALF seems to be lower in CVID than in sarcoidosis (mean, 1.6 vs. 5) [157].

The genetic characterization of a suspected PID is important since the characterization of certain subgroups of PIDs can be accessible to a specific treatment. Functional CTLA4 deficiency, whether due to *CTLA4* haploinsufficiency, *LRBA* deficiency or *DEF6* deficiency, may be suspected in cases with lymphoproliferation (38–73%) and granulomatous disorders (17–45%) associated with complex autoimmunity (mainly cytopenias) and inflammatory enteropathy. This spectrum of diseases is characterized by T-cell lymphopenia, low class-switched B cells and reduced CTLA4 expression on memory Tregs (usually performed with flow cytometry). In *DEF6* deficiency, the T cell exhaustion that is normally present in *CTLA4* and *LRBA* mutated patients is lacking [158]. Infections, including opportunistic ones, affect around 60% of patients [159]. The key to diagnosis is hypogammaglobulinemia, which is present in 50–84% of cases [159,160]. The detection of CTLA4 deficiencies is important because of its associated therapeutic implications since the addition of exogenous CTLA4-Ig (abatacept) can efficiently improve the patients’ condition [160].

Moreover, chronic granulomatous disease (CGD), a primary immune disorder in which phagocytosis is impaired due to pathologic mutation of the *NADPH*, thus inducing an oxidative burst defect, can present with granulomatous features. Even if the X-linked phenotype of CGD is the best-known phenotype, autosomal recessive transmission can occur and as a consequence, the disease can be observed in males and females as well. *Aspergillus* spp. infections as well as *Staphylococcus aureus*, *Klebsiella* spp., *Burkholderia cepacia*, *Serratia marcescens* and *Salmonella* spp. infections are frequent and repeated infectious triggers with impaired phagocytic response leads to granuloma formation around persisting antigens or microorganisms [161,162].

Few other monogenic IEI harboring granulomatous features (either in or out of the CVID spectrum) have been described so far (Table 4). Granulomatous manifestations had already been described in *TNFRSF13B*, *BAFF-R*, *B2M*, *XIAP*, *SH2D1A* and hypomorphic *RAG1/2* mutations [163,164]. PI3K pathway defects (activated PI3 kinase delta syndrome or APDS) had rarely been associated with granulomas. In all described APDS cases in cohorts, granulomas occurred only after BCG (bacillus Calmette et Guérin) vaccination. Of note, not every monogenic CVID phenotypes had been associated with granulomas. This enhances the complex correlation between genotypes and phenotypic classification of IEIs regarding granulomatous manifestations. Nevertheless, the clinician should keep in mind that relevant clinical signs should be looked for when facing granulomatosis such as dysmorphic features, recurrent infections, familial aggregation of cases and hypogammaglobulinemia (which is highly unusual in the course of sarcoidosis) in order to properly diagnose PID.

Among the 26 patients seen in our department, the one described herein presented with mild hypogammaglobulinemia along with a past medical history of recurrent infections. The granulomatous disorder was mostly due to common variable immunodeficiency, and one should note that hypogammaglobulinemia in a context of sarcoidosis is highly unusual and must question the diagnosis of primary humoral immunodeficiency. This highlights the need for careful elimination of differential diagnoses at first visit, especially with serum protein electrophoresis and a meticulous patient interview.

### 3.6. Opportunistic Infections

#### 3.6.1. Case Description

A 52-year-old man was referred to our internal medicine department for investigation of subacute dysarthria and gait disorders in the context of stage IV sarcoidosis in remission for 13 years. His history included cancer of the left kidney, obstructive sleep apnea, gout and post-traumatic blindness. His sarcoidosis was a mediastino pulmonary sarcoidosis considered cured under corticosteroids with calcified adenopathies and pulmonary fibrosis with traction bronchiectasis on the CT scan.

The clinical picture was that of a bilateral static and kinetic cerebellar syndrome predominantly on the left associated with dysarthria evolving for more than 3 months. There was also a left hemiparesis with progressive consciousness disorders and confusion. The rest of the clinical examination was unremarkable. Brain MRI performed at the outset revealed a left cerebellar mass that was not enhanced after gadolinium injection and appeared hypersignal on T2 FLAIR-weighted sequences (Figure 4). The initial biological workup showed no inflammatory syndrome and no alteration of the renal or hepatic balance. The blood count was also normal except for deep lymphopenia with a CD4 count of 167/mm^3^. HIV serology was negative. A first lumbar puncture was considered normal, without hypercellularity, hyperproteinorachia or hypoglycorrhachia. Bacterial and mycobacterial cultures were negative. Herpes PCR (herpes simplex 1, 2 and varicella zoster virus negative) as well as John Cunningham virus (JCV) PCR. There was no evidence of toxoplasmosis or cryptococcosis. At this stage, corticosteroids were introduced for the hypothesis of neurosarcoidosis, without improvement of the patient’s condition. In this context of diagnostic impasse, a brain biopsy was performed. Histological examination showed diffuse inflammation of the white matter with foamy macrophages positive for Periodic Acid Shiff staining and lymphocytes. There was also astrocytic gliosis. The JCV PCR on brain biopsy came back positive as did the one performed on a second lumbar puncture making the diagnosis of progressive multifocal leukoencephalopathy clear.

Treatment with mirtazapine and cidofovir was introduced with good tolerance and allowed a partial regression of the neurological disorders with a favorable evolution thereafter.

#### 3.6.2. Discussion

Sarcoidosis patients may be at risk for opportunistic infections (OIs) and a few cases have been reported so far, even in untreated sarcoidosis patients. It is not clear how OIs in patients with sarcoidosis are different from other groups at risk [8]. Occult primary immunodeficiency, peripheral lymphopenia, deficit in macrophages functions and abnormal autophagy may represent valuable hypotheses to explain this increased risk of OIs in sarcoidosis patients [174]. Nevertheless, corticosteroids and immunosuppressants are the major risk factor for OIs and more widely severe infections in sarcoidosis patients [7,54,175].

Tuberculosis and histoplasmosis have a predominant geographic distribution, with incidence being maximal in areas where these diseases are endemic. Several studies have reported an incidence of 0–10%, but these studies were heterogeneous in terms of purpose, studied populations, sarcoidosis definition, and follow-up periods. One can consider that they should not be properly considered as opportunistic infections since they harbor confounding epidemiological risk factors, like geographic distribution.

All types of OIs have been reported in the setting of sarcoidosis whether they could be of bacterial, viral, parasitic or fungal origin. Cryptococcosis is the most frequently reported infection (59%) followed by mycobacterial (either tuberculous or non-tuberculous) infections (13%), nocardiosis (11%), histoplasmosis (9%), pneumocystosis (9%), and aspergillosis (7%) [8]. Severe OIs, such as progressive multifocal leukoencephalopathy due to JC virus, may also occur and challenge the differential diagnosis of neurosarcoidosis as illustrated by our case. In our case, sarcoidosis flare was suspected, and corticosteroids were introduced, without efficacy. OIs presenting with extrapulmonary features are often misdiagnosed as new localizations of sarcoidosis [54,175]. Chronic pulmonary aspergillosis and especially aspergillomas mostly develop on fibrocystic lungs [176] and patients with sarcoidosis and diffuse or cavitating lung disease present with the chronic pulmonary aspergillosis subtype which is the hallmark of infection on cavitating lung diseases [177].

There is currently no clear explanation for immunodepression in sarcoidosis patients. Even if clear risk factors had already been identified (corticosteroids, immunosuppressants), some patients can present without deep lymphopenia or iatrogenic immunosuppression. One could speculate that defective Treg expansion could be a possible explanation to inefficient immune regulation in sarcoidosis [178].

Of note, the autophagy/mTOR pathway which could be impaired in sarcoidosis also plays an essential role in innate immunity and defense against pathogens. For example, patients with HIV infection have an increased susceptibility to tuberculosis even if CD4+ T-cells are within the normal range. One of the suspected mechanisms to explain this is the autophagy dysregulation by HIV on different cell types [179].

Autophagy is crucial for the elimination of mycobacteria within infected cells and the evasion from the host’s autophagy represents a major virulence factor for intracellular pathogens [180,181]. Non-tuberculous mycobacterial infections were also described during sarcoidosis and impaired macrophage’s functions could be an interesting hypothesis to explain mycobacterial infections in sarcoidosis patients (e.g., defective autophagy, M2 phenotype, defective lysosomal or phagocytic pathways).

Thus, it is currently not possible to conclude an increased risk of OIs during sarcoidosis. In particular, it could not be proven that the CD4+ T-cells count influences the risk of OIs onset. Interestingly, patients who develop cryptococcosis during sarcoidosis’ course were more prone to severe disease (i.e., cardiac and central nervous system sarcoidosis). Thus, many confounding factors can be hypothesized. Patients with severe sarcoidosis could be more prone to be treated with immunosuppressants or long course corticosteroids which increases the risk of infection [7,182]. Lymphopenia and more specifically CD4+ lymphopenia are also associated with active sarcoidosis and the deeper the lymphopenia, the more severe the sarcoidosis [183].

Finally, the occurrence of OI in a patient with sarcoidosis receiving immunosuppressants should prompt a reconsideration of the benefit/risk balance and consideration of a reduction in immunosuppression. Therapies that stimulate cellular immunity (interleukin-2, interleukin-7) have not proven their efficacy and could on the contrary induce sarcoidosis flare. For example, Guffroy et al. recently reported a case of sarcoidosis-associated PML whose OI improves under IL-7 at the cost of a major flare of sarcoidosis with the need for intensive care [184].

This case highlights the possibility of misdiagnosing an opportunistic infection during the course of sarcoidosis. These infections should not be confused with a sarcoidosis flare which would delay the correct diagnosis and correct treatment. The two most described opportunistic infections during sarcoidosis which are PML, and sarcoidosis can easily mimic neurosarcoidosis. Any atypical sign suggesting an alternative diagnosis (such as corticoresistant symptomatology like in our case description) should question a differential diagnosis and especially opportunistic infections, which are rare but deadly when the diagnosis is delayed, especially for PML [178].

### 3.7. Neurosarcoidosis

#### 3.7.1. Case Description

A 30-year-old-female patient was admitted to the neurology department for a one-month history of right hemiparesis associated with perioral paresthesias, gait instability and asthenia. The patient had no treatment other than oral contraception, no particular medical history other than 3 consecutive early miscarriages. She was of Turkish origin and arrived in France 12 years earlier. On admission, she described headaches and presented right arm monoparesis and ataxia with positive Romberg sign without cerebellar or vestibular syndrome. There was no sensory deficit nor cranial nerves involvement nor pyramidal irritation. The remaining physical examination was unremarkable.

The standard biological workup was normal (complete blood cells count, electrolytes, protein C reactive, blood calcium, liver function tests, renal function). There was no vitamin deficiency (B1, B6, B9, B12). Serum protein electrophoresis showed polyclonal hypergammaglobulinemia. ACE was normal, serum lysozyme was increased at 24 mg/L (N < 15 mg/L). Viral serologies were negative for HIV, hepatitis B virus (HBV), hepatitis C virus (HCV), CMV and EBV. Varicella zoster virus (VZV) and toxoplasmosis serologies showed a long-standing immunity with the presence of IgG. The standard immunological work-up was within normal ranges: no anti-nuclear antibodies, ANCA, antiphospholipid antibodies, nor hypocomplementemia. The search for anti-aquaporin 4 antibodies was negative.

The lumbar puncture was normal without pleocytosis (<2 nucleated elements, <100 red blood cells) or increased proteinorrachia. There was no intrathecal synthesis of immunoglobulins. Cerebrospinal fluid (CSF) culture was also negative (standard cultures and mycobacterial cultures). Cerebral Brain MRI showed confluent poorly limited intraaxial pontomesencephalic and cerebellar lesions in T2 and FLAIR hypersignal with low contrast enhancement after gadolinium injection without leptomeningitis (Figure 5).

The ophthalmological evaluation showed bilateral anterior uveitis and intermediate uveitis of the left eye. There were granulomatous retrodescemetic precipitates and vitreous opacities resembling the intermediate uveitis “ant eggs” pattern. The thoracoabdominopelvic CT scan was normal apart from homogeneous hepatomegaly. There was no pathologic lymph node enlargement nor pulmonary parenchymal involvement.

Despite normal minor salivary gland biopsy, medical history was first considered compatible with suggestive of neurosarcoidosis. The patient was treated with three methylprednisolone pulses at 1 g/day followed by a corticosteroid therapy at 1 mg/kg daily for 2 months before considering the decrease. Subsequently, the anti-myelin oligodendrocyte glycoprotein (MOG) antibody test came back positive. The evolution was favorable under corticotherapy with a regression of clinical and radiological lesions. Ophthalmologic examination was normal except for sequelae of uveitis. The final retained diagnostic was MOG antibody-associated disorder (MOGAD) anti-MOG associated inflammatory demyelinating disease. Currently, the patient is going well under prednisone 5 mg/day.

#### 3.7.2. Discussion

MOGAD is usually associated with optic neuritis and/or myelitis [185] but several cases of uveitis have been recently described [186]. In our case, the appearance of the cerebellar peduncle lesion and the absence of intrathecal synthesis of immunoglobulins are highly suggestive of this diagnosis.

CNS demyelinating diseases among which multiple sclerosis (MS), Neuromyelitis Optica-Spectrum Disorders (NMO-SD) and MOGAD represent one of the most important differential diagnoses but many other neurological disorders share common features with neurosarcoidosis (Table 5) [187]. Infectious and neoplastic diseases, lymphoma in particular, must be ruled out as anti-inflammatory drugs or immunosuppressants would be ineffective even detrimental.

Neurosarcoidosis is one of the most difficult organ-site involvement to definitely diagnose, especially because biopsies are difficult to perform in routine practice and/or may represent a significant risk to the patient. Stern et al. recently provided actualized diagnostic criteria for neurosarcoidosis in three distinct categories: (1) definite if granuloma is histologically found on neurologic samples with consistent clinical presentation, (2) probable if granuloma is found on an extra neurologic sample and if the clinical presentation is consistent with sarcoidosis and (3) possible if no granuloma is found [187]. Neurosarcoidosis could affect any part of the nervous system. However, certain clinical and radiological features are particularly suggestive: cranial neuropathy (especially peripheral facial palsy and optic neuritis), leptomeningeal involvement with or without hydrocephalus, hypothalamic and pituitary abnormalities… [188]. It is also important to note that 80 to 90% of patients with neurologic symptoms have, at the same time, ascertainable but often asymptomatic extra neural involvement [187]. In these situations, PET CT is a useful tool to assess both disease activity and potential biopsy sites [187,189]. Thus, isolated neurological damage should prompt the clinician to have a rigorous diagnostic approach to eliminate differential diagnoses. As such, CSF analysis could be very helpful. Neurosarcoidosis is typically associated with lymphocytic pleocytosis and increased CSF protein level. Hypoglycorachia is less frequent (around 15 to 20% of cases) but highly evocative. Immunoglobulins G oligoclonal bands are found in 20 to 50% of neurosarcoidosis patients versus for example more than 95% in MS [190,191]. Other CSF biomarkers like increased IL-6 and CD4/CD8 ratios may help to distinguish neurosarcoidosis from other neurological disorders, MS in particular [192]. As in other forms of sarcoidosis, corticoresistance (a situation when corticosteroids are unable to improve the patient’s condition) should question for a differential diagnosis although it is more frequent in neurosarcoidosis compared with other extrapulmonary organ involvements. In about a quarter of patients, corticosteroid therapy alone is not sufficient or totally ineffective to allow remission [191]. In our case, even if the correct diagnosis was MOGAD, the corticosteroid treatment improved the patient’s condition. In other cases, particularly in lymphoma cases, corticosteroids may improve the patient’s condition to a certain extent but with a substantial risk of deleterious diagnostic delay. Consequently, neurological symptoms developing many years later in patients with already known extra neurologic sarcoidosis should first evoke an alternative diagnosis [190,191].

**Table 5 diagnostics-11-01240-t005:** Main differential diagnoses of central nervous system sarcoidosis.

Diagnostic Subset	Diagnostic Features Shared with Neurosarcoidosis	References
**CNS inflammatory disorders**		
Multiple sclerosis	WM lesions, short myelitis, ON	[192,193,194,195]
NMO-SD and MOGAD	ON especially if bilateral/papilledema, LTEM	[192,196,197]
PACNS	Stroke, leptomeningeal involvement	[192,198]
CLIPPERS	Response to steroids, punctate enhancement	[192,199]
Anti-GFAP associated disorders	Myelitis, meningitis, papilledema, punctate enhancement	[192,200]
**Infections**		
Bacterial: tuberculosis (and other mycobacteria),syphilis, Lyme disease, Whipple disease, brucellosis	Meningitis (hypoglycorrachia), myelitis, ON	[10,192,201]
Fungal: cryptococcosis, histoplasmosis…	Meningitis (hypoglycorrachia)	[54,192,202]
Parasitic: toxoplasmosis, toxocarosis…	Myelitis and ON for toxocarosis, brain mass lesion(s) for toxoplasmosis	[192,203]
**Tumoral conditions**		
Lymphoma	Response to steroids, uveitis	[192,204]
Lymphomatoid granulomatosis	Vasculitis, punctate enhancement	[192,205]
Meningeal carcinomatosis	Cranial nerve involvement, leptomeningeal involvement, hypoglycorrachia	[192]
Metastases	Brain mass or meningeal lesions	[192]
Meningioma	Pachymeningeal involvement	[192]
Histiocytic diseases: Langerhans cell histiocytosis, Erdheim–Chester disease and Rosai–Dorfman syndrome	Hypothalamic-pituitary involvement, brain mass lesions	[90,192,206,207]
Ependymoma	Spinal cord involvement	[192]
Germinoma	Hypothalamic-pituitary or optic nerve involvement	[192]
Glioma	Brain mass lesion(s)	[192]
**Systemic inflammatory disorders**		
Behcet’s disease	Meningitis, uveitis, ON, myelitis	[192,195,208]
GPA	Pachymeningeal involvement, vasculitis	[192,209]
Sjögren syndrome	ON, myelitis	[192,195]
SLE	Myelitis, vasculitis	[192]
IgG4-related pachymeningitis	Pachymeningeal involvement	[187]

Abbreviations: CLIPPERS: chronic lymphocytic inflammation with pontine perivascular enhancement responsive to steroids, CNS: central nervous system, GFAP: glial fibrillary acidic protein, GPA: granulomatosis with polyangiitis, LTEM: longitudinally transverse extensive myelitis, MOGAD: myelin oligodendrocyte glycoprotein antibody-associated disease, NMO-SD: Neuromyelitis Optica-Spectrum Disorders, ON: optic neuritis, PACNS: Primary Angiitis of the CNS, SLE: systemic lupus erythematosus, WM: white matter.

## 4. Discussion

In our tertiary center, among 553 patients referred for a diagnosis of sarcoidosis in the last ten years, 26 (4.7%) have been diagnosed with another condition. These cases, except those described above in the text, are depicted in Table 6.

Alternative diagnoses to granulomatous disorders mimicking sarcoidosis can be separated into seven different groups: (1) infectious disorders (e.g., tuberculosis, cat scratch disease, Whipple’s disease), (2) neoplastic disorders (e.g., Hodgkin’s lymphoma, germinoma), (3) iatrogenic SLR (e.g., SLR due to immune checkpoint blockade, BRAF/MEK inhibitors), (4) device or microparticle-induced granulomatous reactions (e.g., chronic beryllium disease, talc pneumonia, silicosis), (5) primary immunodeficiencies with granulomatous features (e.g., CVID, ataxia telangiectasia), (6) systemic disorders with granulomatous features (e.g., AAV, Blau syndrome, Rosai–Dorfman disease) and (7) some organ specific entities (e.g., hypersensitivity pneumonitis, granuloma annulare, Langerhans cell histiocytosis).

Most of the time, atypical features at diagnosis may help the clinician to distinguish sarcoidosis from its mimickers. Some clinical features are very atypical in a presumed sarcoidosis such as an unusual age at onset (before 25 and after 45 years old except for women for whom a second peak of incidence is noted after 50 years old), impaired general conditions, high fever (can be seen in Löfgren’s syndrome, Heerfordt’s syndrome, hepatic or renal sarcoidosis) clubbing, crackles at auscultation, hemoptysis or acute or subacute dyspnea (especially in the absence of lung fibrosis) [210]. Other organ involvements can be atypical too. For example, digestive tract involvement, isolated ileal involvement is exceptional and must suggest an alternative diagnosis (as we have discussed it for our patient). Corticoresistance is not a classic feature of sarcoidosis and should raise the question of a differential diagnosis (or more simply as a lack of compliance) [3].

In our literature review (Appendix A), corticoresistance was present in 60.2% of the cases. In this review, patients with Whipple’s disease were more prone to present with corticoresistant disease (*p* = 0.02) as well as patients with mycobacterial infections (*p* = 0.02) or primary immunodeficiencies (*p* = 0.003). Exclusive extrathoracic involvement was also more frequently associated with microparticle-driven granulomatosis (*p* = 0.048). Interestingly, patients with infections, solid neoplasia or lymphoma were more prone to heal compared with others (*p* < 0.05). One could assume that this is due to the improvement of therapeutics against infectious agents or neoplasia over the years. To our knowledge, only limited data are available in the medical literature on how medical history, physical examination, laboratory, imaging and pathological investigations can be useful to make the correct diagnosis when facing a sarcoidosis mimicker. The American Thoracic Society recently published an official clinical practice guideline in which a panel of experts discussed and summarized evidence-based diagnostic management of sarcoidosis. Many diseases or pathological conditions may mimic sarcoidosis (Table 1, Table 2, Table 3, Table 4, Table 5 and Table 6).

The clinician must be aware of red flags that may suggest a differential diagnosis whether they would be clinical (as discussed above), radiological (e.g., unilateral hilar lymphadenopathy (3–5% of patients) or exclusive mediastinal lymphadenopathy without hilar lymph node enlargement, compressive lymphadenopathy, anterior mediastinal lymphadenopathy, miliary nodules, ground glass opacities, pleural involvement and bulky or cavitary mass (4%)), biological (e.g., hypogammaglobulinaemia) or histological (extensive or dirty necrosis, palisading granulomas) [6,211,212,213]. Atypical radiological manifestations are of utmost importance. Generally speaking, these atypical presentations should encourage, as illustrated in our exemplary cases, the clinician to pay attention to other causes, including lymphoma, infectious granulomatosis (e.g., tuberculosis, leprosy, syphilis, bartonellosis, brucellosis, Q fever and Whipple’s disease) and common variable immunodeficiency. Drug-related SLR and device or microparticle-induced granulomatous reactions are more easily diagnosed with medical history.

As shown above, the occurrence during the follow-up of atypical organ involvement (e.g., peritoneal or gut involvement) or new organ involvement in a previously controlled sarcoidosis, and refractory disease which is defined as a disease in which a second-line treatment is not sufficient to achieve satisfying disease control or satisfying CS tapering, in a patient with previously known sarcoidosis, must lead to histological confirmation to rule out opportunistic infection or neoplasia and especially lymphoma [3]. Of note, worsening of symptoms under corticosteroids is highly suspicious and should raise the question of a differential diagnosis of infection or neoplasia. When facing suspicion of neurosarcoidosis, PET CT can be useful to determine either CNS inflammation and/or extra neurologic biopsy sites with active sarcoidosis features [214]. Moreover, combination of F18 fluorothymidine and F18-FDG PET CT can be useful to distinguish neoplastic disorders from sarcoidosis [215].

Accordingly, all these unusual circumstances should question the accountability of sarcoidosis and lead the clinician to repeat histologic samples and microbiological analyses. Recent advances brought by translational research provided interesting tools to help the clinician distinguish new entities by making correlations between phenotypes and genotypes. The most striking example concerns the inborn errors of innate immunity. Recent advances in omics techniques especially genomics (whole exome, next generation sequencing, genome wide association studies) allowed a better characterization of primary immunodeficiencies [149]. The Genomic Research in Alpha-1 Antitrypsin Deficiency and Sarcoidosis (GRADS) propose to analyze transcriptome and microbiome data from sarcoidosis patients in order to identify novel biomarkers [216]. In a similar manner, omics techniques have provided interesting diagnostic tools to ease the identification and characterization of mycobacteria, especially matrix-assisted laser desorption ionization-time of flight (MALDI-TOF), mass spectrometry and nucleic acid amplification tests [217].

## 5. Conclusions

Sarcoidosis, being a systemic disease of unknown etiology, can be a real diagnostic challenge. A careful clinical examination along with a rigorous diagnostic approach and an orientated examination of histological samples is of utmost importance in order to assess differential diagnoses. The clinician should be aware of such differentials when facing a granulomatous disorder, especially in case of atypical features for sarcoidosis. The diagnosis of sarcoidosis is based on the elimination of differential diagnoses which makes the diagnostic process all the more important, both for short- and long-term prognosis of the patient.

## Figures and Tables

**Figure 1 diagnostics-11-01240-f001:**
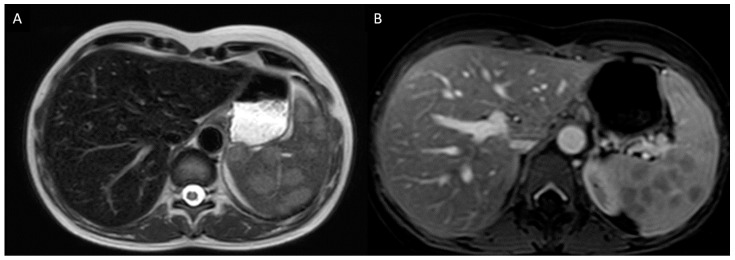
Axial abdominal MRI slices showing hyperintense splenic nodules on T2-weighted sequence (**A**) and hypointense signal on the early gadolinium-enhanced T1 acquisition (**B**).

**Figure 2 diagnostics-11-01240-f002:**
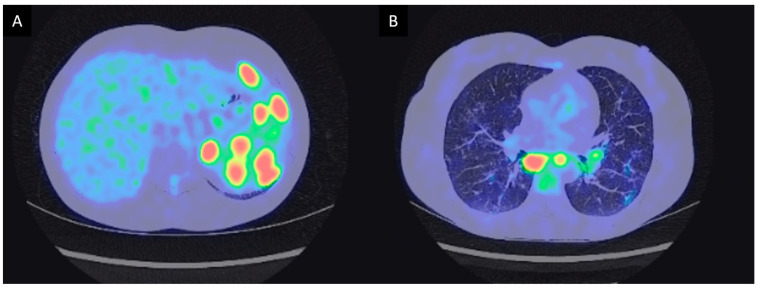
Axial PET-CT acquisitions showing increased metabolic activity in the spleen nodules (**A**) and in mediastinal lymph nodes (**B**).

**Figure 3 diagnostics-11-01240-f003:**
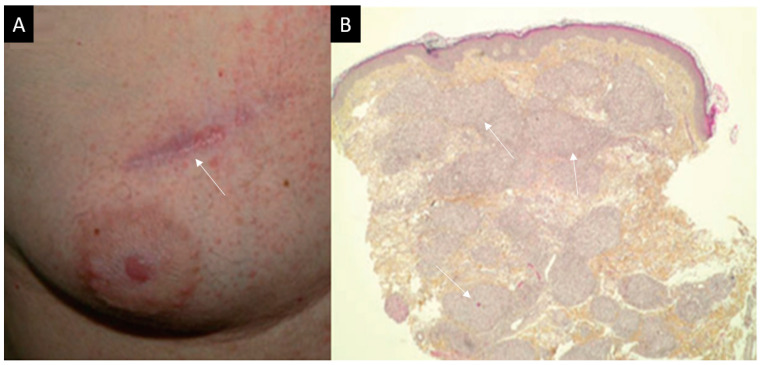
(**A**) Skin nodules on a scar of the anterior chest (white arrow). (**B**) Biopsy specimen of the skin lesion stained with haematoxylin–eosin–saffron showing numerous granulomas (white arrows); original magnification: ×200.

**Figure 4 diagnostics-11-01240-f004:**
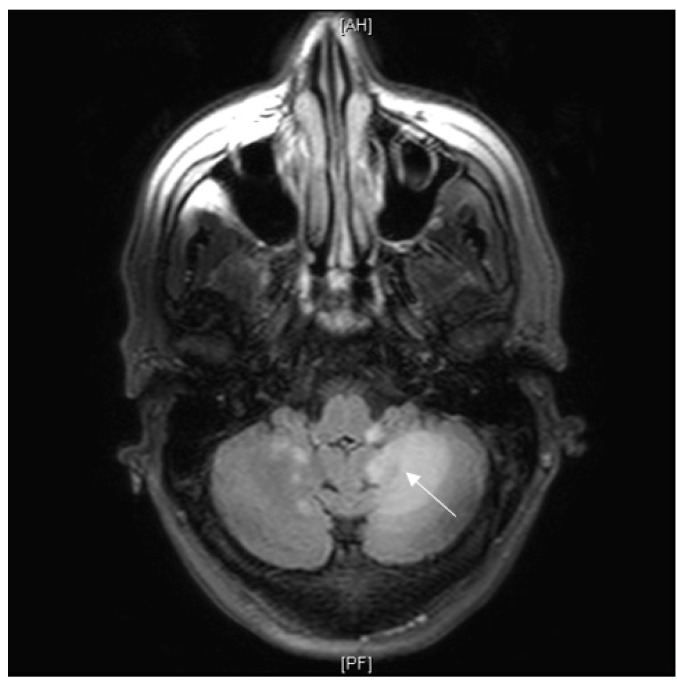
Brain magnetic resonance imaging from our patient with progressive multifocal leukoencephalopathy. Hyperintense left cerebellar mass on T2 FLAIR weighted sequences (arrow).

**Figure 5 diagnostics-11-01240-f005:**
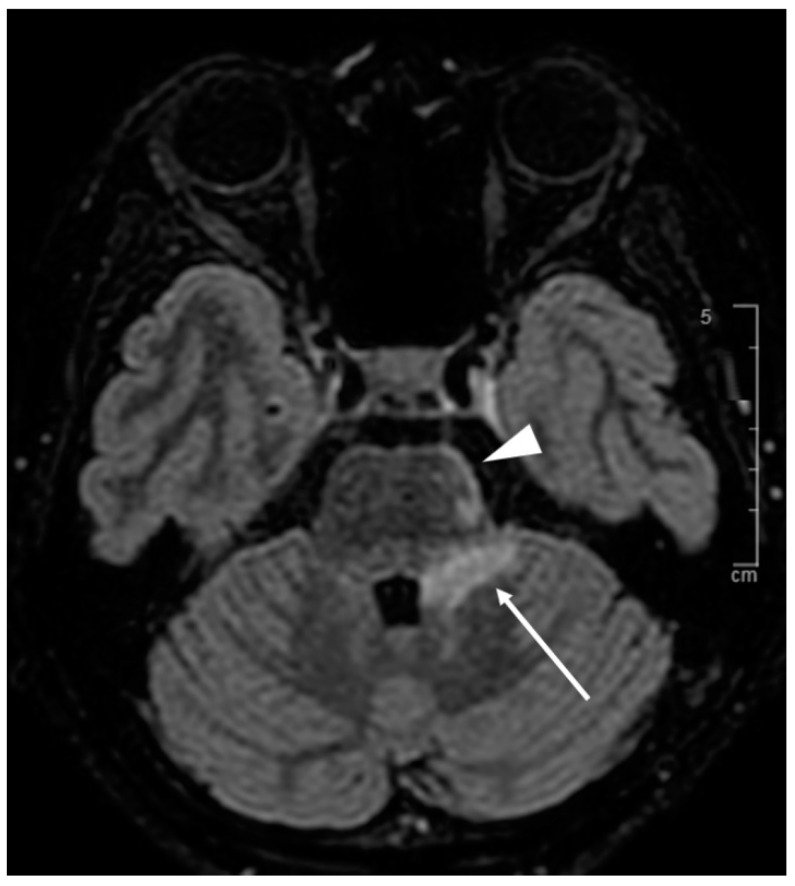
T2 FLAIR weighted sequence of brain magnetic resonance imaging evidencing pontomesencephalic (white arrow head) and right cerebellar inflammatory lesions.

**Table 1 diagnostics-11-01240-t001:** Infectious differential diagnosis of sarcoidosis and their main organ involvement.

Diagnostic Subset	Infectious Agent	Involved Organs	Specific Notes	References
Bacteria				
	*Mycobacterium tuberculosis*	Lung, lymph nodes, eye, skin, liver, arthritis	X	[10]
	*Mycobacterium avium complex* or other atypical mycobacteria	Lung, lymph nodes, skin, eye, digestive tract (*M. bovis*), liver	*M. bovis* is associated with digestive involvement.	[11,12]
	*Bacille de Calmette et Guérin*	Lung, lymph nodes, disseminated	Intravesical for bladder cancer	[13]
	*Mycobacterium leprae*	Skin, PNS, lymph nodes, eye, joint	X	[14]
	*Brucella* spp. (*melitensis*/*abortus*)	Spleen, liver, bone marrow, lymph nodes, arthritis	X	[15]
	*Listeria monocytogenes*	Eye, heart, CNS, granulomatosis infantiseptica in the newborn	Granulomatosis infantiseptica in infants	[16]
	*Salmonella* spp.	Mesenteric lymph nodes, liver, spleen, bone marrow. NB: CGD promotes Salmonella infections	CGD promotes Salmonella infections and infections with other intracellular bacteria (defective oxidative burst)	[17]
	*Nocardia* spp.	Lung, brain, skin, liver	Nocardiosis also occurs in sarcoidosis patients	[18,19]
	*Francisella tularensis*	Lymph nodes, eye, skin, heart, lung	X	[20,21,22]
	*Tropheryma Whipplei*	Joints, CNS, eye, heart, liver, skin, gut	X	[23]
	*Bartonella* spp.	Liver, spleen, skin, lymph nodes, eye, heart	X	[24,25,26,27]
	Q fever (*Coxiella burnetii*)	lung, liver, bone marrow, heart, lymph nodes, arthritis	Granulomas are associated with acute forms, chronic courses of Q fever are more often non granulomatous. Characteristic granuloma in Q fever is called “doughnut granuloma”. Preferentially non necrotizing granulomas.	[28,29]
	Lyme disease	Skin, joint, bone, eye, heart, muscle		[13]
	Syphilis (*Treponema pallidum*)	Skin, eye, lymph nodes, liver, vascular and gummatous syphilis	Granuloma are especially found in tertiary forms (gummatous syphilis and proliferative granuloma)	[30]
	Actinomycosis	Skin, lung, lymph nodes, CNS, digestive tract, liver, pelvis	X	[31]
	Melioidosis (*Burkholderia pseudomallei*)	Lung, skin, joint, bone, CNS, heart	X	[32]
	*Yersinia* spp.	Digestive tract, lymph nodes	X	[33]
	Donovanosis (*K. granulomatosis*)	Lymph nodes, genitals	Context of unsafe sexual practices	[34]
	Lymphogranuloma venereum (*C. trachomatis* serovar L1-L3)	Lymph nodes, genitals, anus	Context of unsafe sexual practices	[35]
Viruses				
	CMV	Any organ	X	[36,37]
	EBV	Lung, skin, lymph nodes (LYG)	X	[38]
	HSV-1, HSV-2 and VZV	Skin, eye, CNS, lung, liver	X	[39,40]
	HIV	Skin (granuloma annulare), any organ (LYG)	Other associated conditions (mycobacteria, CMV, syphilis, …)	[41,42,43]
	HBV	Skin (granuloma annulare), liver	X	[44,45]
	HCV	Skin (granuloma annulare), liver	Possible interferon-induced granulomas	[46,47]
	SARS-CoV2	Skin (granuloma annulare), kidney	X	[48,49,50]
Parasites				
	*Toxoplasma gondii*	CNS, lymph node, eye	X	[13]
	*Schistosoma* spp.	Digestive tract, pelvis, liver	X	[51]
	Leishmaniasis	Skin, spleen, bone marrow, lymph nodes	X	[52]
	*Echinococcus* spp.	Liver, lung	X	[13]
	*Toxocarosis*	Liver, lung, eye, CNS	X	[13]
	*Cysticercosis*	CNS, muscle, eye	X	[53]
Fungi				
	Cryptococcosis	CNS, skin, lymph nodes, lung, joints	Sarcoidosis patients may develop cryptococcosis without immunosuppression	[54]
	Histoplasmosis	Lung, CNS, digestive tract	X	[6,55]
	*Aspergillus* spp.	Lung, skin, disseminated	X	[6,56]
	Blastomycosis	Lung, skin	X	[6]
	*Pneumocystis jiroveci*	Lung	X	[57]

Abbreviations: CGD: chronic granulomatous disease, CNS: central nervous system, EBV: Epstein–Barr virus, HBV: hepatitis B virus, HCV: hepatitis C virus, HIV: human immunodeficiency virus, HSV: herpes simplex virus, PNS: peripheral nervous system, SARS-CoV2: severe acute respiratory syndrome coronavirus type 2, VZV: varicella zoster virus.

**Table 2 diagnostics-11-01240-t002:** Auto inflammatory, autoimmune and systemic conditions mimicking sarcoidosis.

		Ref
Crohn’s disease	Digestive tract, eye, skin, joints, lung	[6]
Granulomatosis with polyangiitis	Lung (consolidations without adenopathies), kidney, sinonasal involvement, peripheral nervous system, skin, eye, joints	
IgG4 related disease	Lymph nodes, pancreas, large vessels, exocrine glands, kidney	[91]
Rheumatoid arthritis	Lung (rheumatoid nodule), joints	[6]
Rosai Dorfman disease	Lymph nodes, skin, CNS	[92,93,94]
Erdeim Chester disease	Pseudo granulomas (any organ)	[95]
Amyloidosis	Any organ	[96,97]
Vogt Koyanagi Harada disease	Eye, CNS, skin	[98]
Blau syndrome	Eye, skin, joints, lymph nodes	[99]
Giant cell arteritis	Vessels, skin	[100]
Autoimmune hepatitis	Liver	[6]
Primary biliary cholangitis		
Primary sclerosing cholangitis		
Kikuchi’s disease	Lymph nodes	[101]
Nieman Pick disease type C	IBD with granulomas	[102]

Abbreviations: CNS: central nervous system; IBD: inflammatory bowel disease.

**Table 3 diagnostics-11-01240-t003:** Drug and device or microparticles-induced sarcoid-like reactions and their main organ involvements.

		Ref
Drug-induced sarcoid-like reactions		
Interferon (alpha or beta)	Lungs, lymph nodes, eyes	[122]
Ribavirin		
Anti-TNF agents (etanercept >adalimumab > infliximab > others)		
Endothelin receptor antagonists (ambrisentan > bosentan > macitentan)		
Checkpoint inhibitors (PD1 and PDL1 antagonists > CTLA4 antagonists)		
BRAF inhibitors/MEK inhibitors		
Tocilizumab		
Brentuximab vedotin		
Microparticles-induced sarcoid-like reactions		
Chronic beryllium disease	Lung, bone marrow, liver, lymph nodes, heart, skin	[123]
Aluminium	Lung, digestive tract, injection sites	[124,125]
Silicosis	Skin, lymph nodes	[126,127,128,129]
Talc	Skin, liver, lung	[130,131,132]
Coal	Bone marrow, local exposure	[133,134]
Silicone	Breast (local exposure), disseminated if implant rupture	[135,136]

Abbreviations CTLA4: cytotoxic T-lymphocyte associated protein 4; PD1/PDL1: programmed cell death (ligand) 1; TNF: tumor necrosis factor.

**Table 4 diagnostics-11-01240-t004:** Primary immunodeficiencies with granulomatous features.

		Ref
CVID		
LRBA deficiency	GLILD	[158]
CTLA4 haploinsufficiency		
Other primary immune deficiencies		
Chronic granulomatous disease	Skin, lung, sinuses	[165]
RAG1-2 deficiency	Skin, lung, sinuses (GPA-like)	[166]
TAP1/2 and TAPBP deficiency	Skin	[167]
PLAID (PLCG2)	Skin	[99,168]
Ataxia Telangiectasia	Skin, joints, bones	[169]
NAID	Digestive tract, joints, skin, eyes	[99]
XLP1-2	LYG	[170]
Hermansky Pudlak syndrome	Digestive tract, skin	[171,172]
CARD9 deficiency	Skin (dermatophytosis)	[173]
XIAP deficiency	Lymph nodes, digestive tract	[102]

Abbreviations: CARD9: Caspase recruitment domain-containing protein 9, CTLA4: cytotoxic T-lymphocyte associated protein 4, CVID: common variable immunodeficiency, GLILD: granulomatous lymphocytic interstitial lung disease, GPA: granulomatosis with polyangiitis, LRBA: LPS-responsive beige-like anchor protein, LYG: lymphomatoid granulomatosis, NAID: NOD associated autoinflammatory disorder, PLAID: PLCG2 auto inflammatory disorder, RAG: recombinant activating gene, TAP: transporter for antigen presentation/BP: binding protein, XIAP: X-linked inhibitor of apoptosis protein, XLP: X-linked lymphoproliferative disease.

**Table 6 diagnostics-11-01240-t006:** Remaining cases with a wrong initial diagnosis of sarcoidosis in our tertiary center.

Case	Sex	Age	Ethnicity	Clinical Features	Atypical Features for Sarcoidosis	Time to Correct Diagnosis	Diagnosis	Treatment	Outcome
1 *	M	56	Caucasian	Hemoptysis, mediastinal lymph nodes, multiple parenchymal condensations with excavations	Hemoptysis, excavated parenchymal condensation on chest CT	4 weeks	Tularemia	21 days doxycyclin course	Healing
2	F	55	Caucasian	Isolated necrotic left sus clavicular adenopathy with hepatomegaly. Heterogenous liver echogenicity	Isolated extramediastinal lymph node without parenchymal involvement	5 months	Cat scratch disease	1 month course doxycyclin	Healing
3	M	45	Maghrebian	Altered general condition with 2 kg weight loss. Hard, fixed and painless left sus clavicular and cervical adenopathies. Sino nasal obstruction with retropharyngeal adenopathies with cavum mucosal thickening	Atypical lymphadenopathies, exclusive extrathoracic multi organ involvement.	24 months	Hodgkin’s lymphoma	Chemotherapy (ABVD, ICE)	Currently continuing chemotherapy
4	F	56	Caucasian	Cervical and axilar lymph nodes. Cavum tumefaction. Granulomas without atypical features. Atypical CD30+ cells on repeated lymph nodes sampling.	Cavum infiltration and exclusive extrathoracic lymph nodes.	72 months	Biclonal lymphoma (Hodgkin and Diffuse large B cell lymphoma). Hodgkin disease was already present at disease onset 6 years before (second-look histological examination).	Chemotherapy (R-CHOP 8x)	Healing
5	M	54	Maghrebian	Lower esophagus stenosis with peri esophageal adenopathies and dysphagia. Paratracheal and subcarinal and antero superior mediastinal lymph nodes. Histological examination concordant with Piringer Kuchinka’s lymphadenitis.	Compressive phenomenon. No hilar lymph nodes with anterior mediastinal lymph nodes.	40 months	EBV positive Hodgkin’s lymphoma	ABVD 6 courses	Healing
6	M	75	Hispanic	Compressive right orbital infiltrate. Isolated enlarged lymph node of Barety area.	Compressive phenomenon. Isolated mediastinal lymph node without hilar lymph node.	17 months	Right oribtal marginal zone lymphoma (previously improved by local corticosteroids more than a year before for suspected scleritis).	Surgical resection.	Healing
7*	F	63	Caucasian	Bilateral anterior and intermediate uveitis. Granuloma on MSGB. Gait disturbance with multiple supra tentorial demyelinating lesions on FLAIR-weighted sequences.	Corticoresistant uveitis and neurological involvement.	3 years	Vitroretinal lymphoma.	R-Metho AraC chemotherapy followed by ibrutinib and R-VP16-Holoxan.	The patient died 3 months after the diagnosis [204].
8	M	53	Caucasian	Granulomatous kidney (renal failure) and liver disease (cirrhosis and portal hypertension). Mesenteric and cervical lymph nodes. Monoclonal gammopathy.	Exclusive extrathoracic disease with severe renal involvement.	47 months	Multiple myeloma	Granulomatosis was treated with corticosteroids, azathioprine and mycophenolate mofetil without clear improvement.	The patient died a few days after trans jugular portal shunt procedure.
9	M	50	Caucasian	Sus and subdiaphragmatic lymph nodes in the course of rheumatoid arthritis.	Ground glass opacities and compressive lymph nodes.	NA	Sarcoid like reaction to ETN (at introduction)	ETN withdrawal. Monoclonal antibody to TNFa did not provoke SLR recurrence. Rheumatoid arthritis and SLR improved under ustekinumab.	Healing
10*	F	59	Caucasian	Bilateral anterior and intermediate uveitis mediastinal lymphadenopathy. Previously treated with ADA for rheumatoid arthritis.	No atypical features	Concomittant	SLR to ADA	ADA withdrawal, Switch to tofacitinib + MTX + CS	Free of symptoms under tofacitinib and MTX.
11	M	80	Caucasian	Bilateral intermediate uveitis and mediastinal lymphadenopathies	No atypical features	Concomittant	SLR to ETN	Switch to ADA	Free of symptoms under ADA.
12	F	38	Caucasian	Uveitis, sarcoids. Lung parenchymal involvement and mediastino hilar lymphadenopathies. Melanoma with vemurafenib and cobimetinib treatment.	No atypical features.	Concomittant	SLR to vemurafenib and cobimetinib (9 months exposure)	CS	The patient died of her melanoma without severe organ involvement due to SLR. CS improved SLR.
13	F	56	Caucasian	Bilateral panuveitis and sarcoids.	No atypical features.	NA	SLR to ADA (no exposure data).	Local CS and ADA withdrawal.	Healing without recurrence.
14	M	59	Caucasian	Parenchymal lung involvement, hepatosplenomegaly, bone marrow failure. Renal failure.	Febrile pancytopenia	Concomittant	Mycobacterium genavense in a previously known sarcoidosis.	Ansatipine, clarithromycin and moxifloxacin	Healing.
15	M	18	Maghrebian	Liver, spleen, lung and bone marrow involvement. Diffusely enlarged lymph nodes;	Early onset (18 years old), past medical history of opportunistic infections (actinomycosis) and hypogammaglobulinemia	180 months	LOCID	CS and IVIg	Stable pulmonary function under CS. No infection under IVIg.
16	F	37	Caucasian	Hypercalcemia, parenchymal lung involvement with mediastino hilar lymph nodes. Skin granulomas.	Repeated pulmonary infections and hypogammaglobulinemia.	264 months	CVID-RGD	CS	Healing. The patient remained free from infections under IVIg.

Abbreviations: ABVD: adriamycin-bleomycin-vincristine-dexamethasone, ADA: adalimumab, AraC: aracytine, ChlVPP: Chlorambucil-Vinblastine-Procarbazine-Prednisone, CS: corticosteroids, CT: computed tomography, CVID: common variable immunodeficiency, ETN: etanercept, SLR: sarcoid-like reaction, IFX: infliximab, IVIg: intravenous immunoglobulins, MTX: methotrexate, NA: not available, NMOSD: neuromyelitis optica spectrum disorder, RGD: related granulomatous disorder, R-CHOP: rituximab-cyclophosphamide-hydroxydoxorubicin-oncovin (vincristine)-prednisone,* already published cases.

## Data Availability

The dataset used for statistical analysis is cited as Appendix A.

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
