# Peer review of "Challenging Mimickers in the Diagnosis of Sarcoidosis: A Case Study"

_diagnostics, 2021, doi:10.3390/diagnostics11071240_

Round 1

Reviewer 1 Report

Please see the Word document attached

Author Response

Response to reviewer 1 :

We thank the reviewer for his/her comments regarding our manuscript.

Q: Page 2, line 83: When does this trip occurred in months (rather than a date) before first consultation? Same suggestion for line 85

A: We agree that a more precise timeline is needed. Please find the modifications on page 3, lines 83 and 90.

“He reported a trip to India four months after the onset of symptoms, with no medical issues at the time.”

“Of note, a few days after returning from India (a trip the patient took after the onset of oral ulceration), he developed dry cough which led to further investigations”

Q: Page 7, I think the paragraphs including the IGRA and sputum regarding TB diagnosis may be a little off topic. I understand the authors want to point the superiority of QFT-GIT, it less obvious concerning the PCR / Xpert. I suggest to shorten this paragraph or take it off.

A: We understand that this paragraph can be considered as off topic. We have decided to shorten it by deleting the end of the paragraph since precisions about the different IGRA tests available. Please find the modifications on page 8.

Q: Page 8: Please explain why a medullogram was performed. During bronchoscopy, please specify if bronchial biopsies were obtained and also results for TB.

A: A medullogram was performed in the context of a diagnostic procedure for unexplained inflammatory syndrome. We have precised this statement with the following sentence on page 9, line 226: “Since the first line investigations remained undiagnostic, a bone marrow aspiration was performed”

The probabilistic doxycycline treatment was administered in the hypothesis of an infection with intracellular bacteria, since pyoeopithelioid inflammation was found on histological samples.

The precision was made line 235: “In case of occult infection with intracellular bacteria”

Q: Page 11: please specify the sample type for TB

A: the sample for TB was a digestive sample collected during an endoscopic procedure. This statement was precised on line 324 [intestine samples].

Q : Page 12 Please specify “CD” which I assumed stands for Cronh disease

A: This abbreviation was defined at the same place in the text.

Q: Page 14: Please reformulate the paragraph on BS and NOD2 which presently seems a bit off topic

A: We agree that this way, the paragraph concerning Blau syndrome could appear a bit off topic. We modified it by adding this sentence at the beginning of the paragraph: “Other diseases sharing the same impaired NOD2 pathway background may mimick sarcoidosis” (line 421). We hope that this transition is clearer in that way.

Q: Page 15: Case presentation: Please specify if other organs were involved (especially lungs). Colonic perforation etiology?

A: There were no lung parenchymal involvement at this time. The colonic perforation was spontaneous, assumed to be due to secondary digestive localization of its primary tumor. Please find the modifications on page 16, lines 454 and 461. “There was no parenchymal lung involvement on chest CT. The etiologic workup of this uveitis excluded other secondary causes of uveitis. Corticosteroids eyedrops were suc-cessfully introduced. However, the uveitis became chronic with multiple flares at top-ical corticosteroids withdrawal, and oral corticosteroids were therefore initiated in combination with hydroxychloroquine. While the uveitis and cutaneous granulomatosis were controlled under this treatment, the patient experienced a relapse of his metastatic melanoma. Unfortunately, he died of colonic perforation due to secondary localizations of his melanoma despite the use of several unsuccessful therapeutic lines.”

Q: Page 20, Reference 164 is noted twice. Concerning CGD, the below reference could be of interest, concerning infectious and non-infectious complications of CGD: “Salvator H. et al Pulmonary manifestations in adult patients with chronic granulomatous disease”

A: We thank the reviewer for noticing it. The reference 164 was corrected and the proposed reference, which is very interesting, was added. Please find the modifications at the same place in the text.

Q: Concerning the case report (primary immune deficiencies); is there any information regarding outcome disease and patient evolution?

A: We have provided precisions about the outcome of the disease. The patient remained well under immunoglobulin substitution. Please find the modifications on line 586 : “The patient remain well under immunoglobulin substitution.”

Q: Regarding the paragraph dealing with opportunistic infections, I would say it could be a little bit off topic. Indeed, the title is “challenging mimickers in sarcoidosis diagnosis”. Here is presented a case and are discussed cases of difficult diagnosis of infection. I understand the global idea of the authors to highlight that all new symptoms in sarcoidosis may not be sarcoidosis flare, however would suggest to cancel this case and discussion since the paper is already quite long.

If the authors disagree with this suggestion, I wouldn’t be totally opposed but suggest to shift it as the last case of the 7 and rewrite the paragraph with a better presentation of OI causes and better presentation of epidemiology of such OI.

A: We agree that the section concerning opportunistic infections in sarcoidosis may appear to be a little bit off topic. We assume that opportunistic infections, which are rare but unexceptional in sarcoidosis patients, are diagnostic pitfalls to keep in mind when facing unusually corticoresistant sarcoidosis flare. We have deleted the two paragraphs related to infection prophylaxis which was clearly out of scope and clarified the statement about opportunistic infections misdiagnosed as sarcoidosis flare. Please find the modifications from line 793 to line 800 “This case highlights the possibility of misdiagnosing an opportunistic infection during the course of sarcoidosis. These infections should not be confused with a sarcoidosis flare which would delay the correct diagnosis and correct treatment. The two most described opportunistic infections during sarcoidosis which are PML and sarcoidosis can easily mimick neurosarcoidosis. Any atypical sign suggesting an alternative diagnosis (such as corticoresistant symptomatology like in our case description) should question a differential diagnosis and especially opportunistic infections, which are rare but deadly when the diagnosis is delayed, especially for PML [178]”

Q: Page 23 line 742, They, instead of the

A: This typo was corrected at the same place in the text.

Q: Last page: Could the authors provide a conclusion

A: We agree that a valuable conclusion could be provided.

Please find the modification at the very end of the manuscript:

“5. Conclusion

Sarcoidosis being a systemic disease of unknown etiology can be a real diagnostic challenge. A careful clinical examination along with a rigorous diagnostic approach and an orientated examination of histological samples is of utmost importance in order to assess differential diagnoses. The clinician should be aware of such differentials when facing a granulomatous disorder, especially in case of atypical features for sarcoidosis. The diagnosis of sarcoidosis is based on the elimination of differential diagnoses which makes the diagnostic process all the more important, both for the short and long term prognosis of the patient.”

Reviewer 2 Report

There is no doubt that this contribution offers a practical, valuable and meaningful tool for both experts clinicians and resident physicians. I have no specific suggestions or any further comment to add.

Author Response

Response to reviewer 2:

There is no doubt that this contribution offers a practical, valuable and meaningful tool for both experts clinicians and resident physicians. I have no specific suggestions or any further comment to add.

We thank the reviewer for his/her careful examination and remarks about our manuscript.

Reviewer 3 Report

General comments:

The paper is complete and well written, however, there are many imprecisions in the text, therefore a linguistic revision is recommended. Please find more detailed comments below.

Major Strengths:

*     Sarcoidosis is an interesting topic, especially regarding differential diagnosis, because of its variety of presentations;

*     Very interesting cases presented.

Major Weaknesses:

*     Captions of Figure 1 and Figure 5 should be more detailed, lack the information about sequences used and description of lesions;

*     There are many errors in the text:

*     Cases presentation – Infectious disease (pg 3, line 100): “resonnance” should be “resonance”

*     Cases presentation – Infectious disease (pg 3, line 106): lack of space after “performed.”

*     Cases presentation – Infectious disease (pg 7, line 154): “non neglictable” should be “non neglectable”

*     Cases presentation – Infectious disease (pg 7, line 158): “quiantifies should be “quantifies”.

*     Cases presentation – Infectious disease (pg 7, line 171): “rapid moleculare tests” should be “rapid molecular tests”

*     Cases presentation – Infectious disease (pg 7, line 182): “dervied antigens” should be “derived antigens”

*     Cases presentation – Neoplastic disorders (pg 10, line 291): “inflamatory” should be “inflammatory”

*     Cases presentation – Neoplastic disorders (pg 10, line 296): “neded” should be “needed”

*     Cases presentation – Inflammatory disorders and systemic diseases (pg 12, line 365): “asociated vasculatides” should be “associated vasculitides”

*     Cases presentation – Inflammatory disorders and systemic diseases (pg 14, line 411): “wieght loss” should be “weight loss”

*     Cases presentation – Inflammatory disorders and systemic diseases (pg 14, line 433): “sinc” should be “since”

*     Cases presentation – Drug-implant and device-induced sarcoidosis (pg 14, line 411): “wieght loss” should be “weight loss”

*     Cases presentation – Drug-implant and device-induced sarcoidosis (pg 17, line 489): “concomittantly” should be “concomitantly”

*     Cases presentation – Primary immune deficiencies (pg 18, line 579): “transitionnal B cells” should be “transitional B cells”

*     Cases presentation – Primary immune deficiencies (pg 21, line 685): “hypogammaglobulineami” should be “hypogammaglobulinemia”

*     Cases presentation – Opportunistic infections (pg 23, line 758): “immunosuppresion” should be “immunosuppression”

*     Cases presentation – Neurosarcoidosis (pg 25, line 838): “cerebellous lesions” should be “cerebellar lesions”

*     Cases presentation – Neurosarcoidosis (pg 25, line 842): “ressembling” should be “resembling”

*     Cases presentation – Neurosarcoidosis (pg 25, line 843): “enlargment” should be “enlargement”

*     Cases presentation – Neurosarcoidosis (pg 25, line 847): lack of space between with and suggestive

*     Cases presentation – Neurosarcoidosis (pg 25, line 852): “sequelea” should be “sequelae”

*     pg 33, line 927: “dinstinguish” should be “distinguish”

Author Response

Response to reviewer 3:

General comments:

The paper is complete and well written, however, there are many imprecisions in the text, therefore a linguistic revision is recommended. Please find more detailed comments below.

We thank the reviewer for this comment and for the consideration for the manuscript.

Major Strengths:

*     Sarcoidosis is an interesting topic, especially regarding differential diagnosis, because of its variety of presentations;

*     Very interesting cases presented.

Major Weaknesses:

Q: Captions of Figure 1 and Figure 5 should be more detailed, lack the information about sequences used and description of lesions;

A: The MRI sequences have been precised. Please find the modifications in the main manuscript under figure 1 and 5.

*     There are many errors in the text:

*     Cases presentation – Infectious disease (pg 3, line 100): “resonnance” should be “resonance”
*     Cases presentation – Infectious disease (pg 3, line 106): lack of space after “performed.”

*     Cases presentation – Infectious disease (pg 7, line 154): “non neglictable” should be “non neglectable”

*     Cases presentation – Infectious disease (pg 7, line 158): “quiantifies should be “quantifies”.

*     Cases presentation – Infectious disease (pg 7, line 171): “rapid moleculare tests” should be “rapid molecular tests”

*     Cases presentation – Infectious disease (pg 7, line 182): “dervied antigens” should be “derived antigens”

*     Cases presentation – Neoplastic disorders (pg 10, line 291): “inflamatory” should be “inflammatory”

*     Cases presentation – Neoplastic disorders (pg 10, line 296): “neded” should be “needed”

*     Cases presentation – Inflammatory disorders and systemic diseases (pg 12, line 365): “asociated vasculatides” should be “associated vasculitides”

*     Cases presentation – Inflammatory disorders and systemic diseases (pg 14, line 411): “wieght loss” should be “weight loss”

*     Cases presentation – Inflammatory disorders and systemic diseases (pg 14, line 433): “sinc” should be “since”

*     Cases presentation – Drug-implant and device-induced sarcoidosis (pg 14, line 411): “wieght loss” should be “weight loss”

*     Cases presentation – Drug-implant and device-induced sarcoidosis (pg 17, line 489): “concomittantly” should be “concomitantly”

*     Cases presentation – Primary immune deficiencies (pg 18, line 579): “transitionnal B cells” should be “transitional B cells”

*     Cases presentation – Primary immune deficiencies (pg 21, line 685): “hypogammaglobulineami” should be “hypogammaglobulinemia”

*     Cases presentation – Opportunistic infections (pg 23, line 758): “immunosuppresion” should be “immunosuppression”

*     Cases presentation – Neurosarcoidosis (pg 25, line 838): “cerebellous lesions” should be “cerebellar lesions”

*     Cases presentation – Neurosarcoidosis (pg 25, line 842): “ressembling” should be “resembling”

*     Cases presentation – Neurosarcoidosis (pg 25, line 843): “enlargment” should be “enlargement”

*     Cases presentation – Neurosarcoidosis (pg 25, line 847): lack of space between with and suggestive

*     Cases presentation – Neurosarcoidosis (pg 25, line 852): “sequelea” should be “sequelae”

*     pg 33, line 927: “dinstinguish” should be “distinguish”

A: We thank the reviewer for his/her attentive reading. All theses typo’s have been corrected at the same place in the text.